# WaveSSM: Multiscale State-Space Models for Non-stationary Signal Attention

Ruben Solozabal [* 1]   Velibor Bojkovic [* 1]   Hilal Alquabeh [1 2]   Klea Ziu [1]   Kentaro Inui [1 2]   Martin Takáč [1]

## Abstract

State-space models (SSMs) have emerged as a powerful foundation for long-range sequence modeling, with the HiPPO framework showing that continuous-time projection operators can be used to derive stable, memory-efficient dynamical systems that encode the past history of the input signal. However, existing projection-based SSMs often rely on polynomial bases with global temporal support, whose inductive biases are poorly matched to signals exhibiting localized or transient structure. In this work, we introduce *WaveSSM*, a collection of SSMs constructed over wavelet frames. Our key observation is that wavelet frames yield a localized support on the temporal dimension, useful for tasks requiring precise localization. Empirically, we show that on equal conditions, *WaveSSM* outperforms orthogonal counterparts as S4 on real-world datasets with transient dynamics, including physiological signals on the PTB-XL dataset and raw audio on Speech Commands.

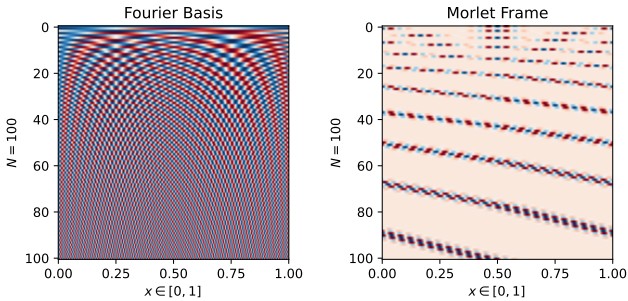

*Figure 1.* Comparison between global polynomial bases and localized wavelet frames. **Left:** Legendre orthogonal-basis functions are utilized in the HiPPO framework to construct an online projection of the input signal with global support. **Right:** wavelet-based frames (Morlet in the example) exhibit more localized time support, promoting selective representations with the ability to attend transient events more effectively.

## 1. Introduction

Modeling long-range dependencies in sequential data remains one of the central challenges in modern machine learning. From natural language, speech processing, or analyzing sensory data, effective sequence models must capture both long-range temporal structure and localized transient events while being computationally efficient. Classical recurrent neural networks (RNNs) and their gated variants (e.g., LSTMs (Hochreiter & Schmidhuber, 1997)) are limited by vanishing and exploding gradients, while attention-based Transformers face quadratic scaling with sequence length. Recently, *state-space models* (SSMs), such as HiPPO (Gu et al., 2020), S4 (Gu et al., 2022b), and Mamba (Gu & Dao, 2023), have emerged as powerful al-

ternatives, providing theoretically grounded, sub-quadratic mechanisms for continuous-time sequence representation.

Despite their success, existing SSM formulations share an important limitation: they rely on a family of *orthogonal polynomials* (Gu et al., 2020; 2023) (e.g., Legendre, Laguerre, or Chebyshev) for constructing the transition dynamics. These bases have global support, meaning each basis element spans the entire signal temporal domain, and therefore cannot efficiently represent non-stationary or localized patterns that vary across the time-scale. As a result, conventional HiPPO-style SSMs tend to fail to adaptively focus on regions of interest in the input sequences.

In this work, our goal is to endow state-space sequence models with an explicitly *multiscale and time-localized* representation. SSMs constructed using orthogonal-basis (i.e., HiPPO-based models (Gu et al., 2022b; Smith et al., 2022; Orvieto et al., 2023)) are powerful for globally smooth signal representation, but their states are inherently global: a single coefficient typically mixes information from the entire history, which can be inefficient for signals with localized events, sharp transitions, or piecewise-smooth structure. Motivated by the approximation-theoretic advantage of wavelets on such data, we introduce *WaveSSM*, a wavelet-based state-space model whose latent states correspond to localized multiscale atoms that can therefore represent and propagate information at different temporal resolutions.

---

[*]Equal contribution  [1]MBZUAI [2]RIKEN AIP. Correspondence to: Ruben Solozabal <ruben.solozabal@mbzuai.ac.ae>.

*Proceedings of the 43rd International Conference on Machine Learning*, Seoul, South Korea. PMLR 306, 2026. Copyright 2026 by the author(s).

We show that WaveSSM is particularly advantageous for modalities such as physiological signals, time series, and audio streams that exhibit *non-stationarity* and *transient* structure. We demonstrate empirically that our WaveSSM model achieves superior accuracy on several benchmark datasets, outperforming SSMs based on orthogonal bases when tested on equal conditions. Our main contributions are as follows:

1. **Wavelet-induced SSMs.** We derive a principled construction of SSMs dynamics from continuous and discrete wavelet frames. The main property of our formulation is that it induces *temporally localized* state coordinates, i.e., information from different temporal regions is stored in different subsets of the state.

2. **Analysis of the stability considerations.** We characterize the numerical stability challenges introduced by wavelet-induced dynamics and discuss design choices that improve stability and yield reliable long-horizon kernels.

3. **Empirical validation on relevant benchmarks.** We evaluate WaveSSM on a suite of real-world tasks emphasizing transient and non-stationary behavior (e.g., PTB-XL, Speech Commands), but also more general benchmarks as Long Range Arena (LRA), demonstrating its gains over orthogonal-basis SSM counterparts under controlled comparisons.

## 2. Background

### 2.1. State-Space Models for Sequence Representation

Continuous-time SSMs provide an expressive and computationally efficient framework for representing long-term dependencies in sequential data. An SSM describes the evolution of a hidden state $h(t)$ driven by an input signal $u(t)$ normally through a linear time-invariant (LTI) system. Such LTI-SSM is defined by

$$\dot{h}(t) = A\,h(t) + B\,u(t), \qquad y(t) = C\,h(t), \quad (1)$$

where $u(t)$ is an input signal, $h(t) \in \mathbb{R}^N$ is a latent state, and $A, B, C$ are learned or designed system matrices. After discretization with step size $\Delta$, one obtains the recurrence

$$h_{t+1} = \bar{A}\,h_t + \bar{B}\,u_t, \quad (2)$$

with $\bar{A}, \bar{B}$ derived from a numerical integrator such as the bilinear transform. This produces a linear time-invariant convolution kernel $K = (C\bar{A}^l\bar{B})_{l \geq 0}$, allowing SSMs to compute long-range dependencies with $O(N)$ memory and $O(L \log L)$ parallel kernel evaluations.

To use SSMs as sequence models, one typically chooses the latent state $h(t)$ to encode a summary of the input history $\{u(\tau) : \tau \leq t\}$. Classical approaches (e.g. Liquid Time-Constant networks (Hasani et al., 2021), Neural CDEs (Kidger et al., 2020), and modern S4 layers (Gu et al., 2022b; Smith et al., 2023)) differ primarily in how they choose or learn the matrices $A$ and $B$ and how these matrices govern the decay and mixing of information over time.

**Projection-based SSMs.** In this paper, we focus on a successful family of SSMs that builds $h(t)$ as an *online projection* of the recent signal history onto a set of basis functions. Let $\{\phi_n\}$ be such functions. Define

$$h_n(t) = \int_0^t u(\tau)\,\phi_n(\tau, t)\,d\tau, \quad (3)$$

where $\phi_n(\tau, t)$ may depend on $t$ through a measure function. Differentiating $h(t)$ yields an ODE of the form (1) whose matrices $A$ and $B$ are determined by the structure of the basis. This idea underlies the HiPPO and SaFARi frameworks.

**The HiPPO framework.** The High-Order Polynomial Projection Operator (HiPPO) framework (Gu et al., 2020) defines an online procedure for projecting the history of the input signal onto a polynomial basis. At time $t$, let $\{P_n(\cdot, t)\}_{n \geq 0}$ denote the family of orthonormal polynomials associated with a time-varying measure $\mu_t$ (e.g. scaled or translated). The HiPPO coefficients are defined as the optimal projection

$$h_n(t) = \langle u_{[0,t]}, P_n(\cdot, t)\rangle_{\mu_t} = \int_0^t u(\tau)\,P_n(\tau, t)\,\mu_t(\tau)\,d\tau, \quad (4)$$

which represent an online summary of the history of $u_{[0,t]}$.

Different measures produce different memory behaviors. The *scaled* measure, $\mu_{\text{sc}}(\tau) = \frac{1}{t}\mathbf{1}_{\tau \in [0,t]}$, distributes weight uniformly over all past times and leads to global representations. Whereas the *translated* measure, $\mu_{\text{tr}}(\tau) = \frac{1}{\theta}\mathbf{1}_{\tau \in [t-\theta,t]}$, restricts memory to a sliding window of length $\theta$. While these measures allow trade-offs between global context and local recency, existing HiPPO formulations lack an explicit mechanism for selectively accessing specific temporal regions of the input.

**The SaFARi framework.** The State-Space Models for Frame-Agnostic Representation or SaFARi (Babaei et al., 2025) generalizes HiPPO by providing a numerical procedure to derive SSM parameters $(A, B)$ from *any* frame or basis, not only orthogonal polynomials. Given a frame $\Phi = \{\varphi_n\}$ in a Hilbert space with its dual $\tilde{\Phi}$, SaFARi defines the representation coefficients

$$h_n(T) = \int_0^T u(t)\varphi_n^{(T)}(t)\mu(t)dt, \quad (5)$$

and shows that their temporal evolution again follows a first-order SSM where $A$ and $B$ are obtained from inner

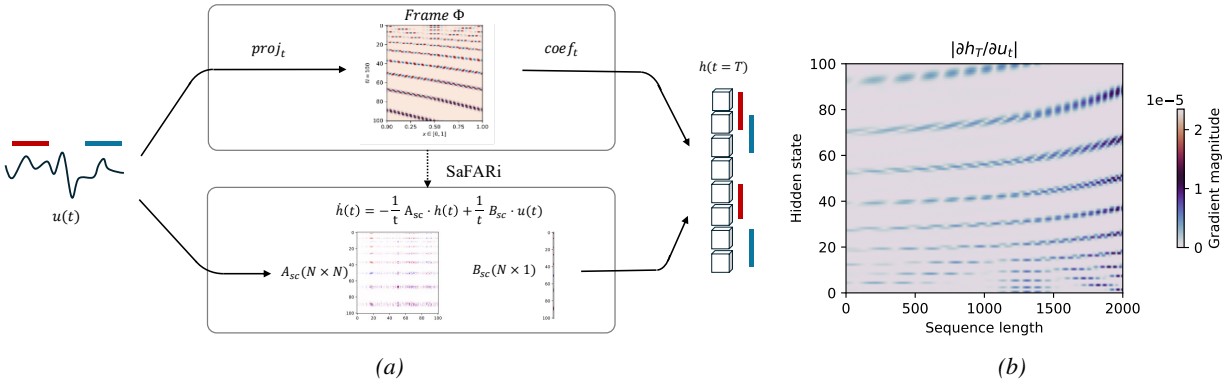

 

*Figure 2.* **Left:** Overview of the WaveSSM framework. The input signal is projected onto a wavelet frame, inducing a continuous-time state-space representation via differentiation of the projection. The resulting SSM maintains a compact latent state with addressable, time-local components, which can be decoded independently or processed for sequence modeling. **Right:** Visualization of the Jacobian $\partial h_T / \partial x_t$, illustrating the influence of inputs at different time steps on the final hidden state. Results are shown for Morlet wavelets of order $N = 100$ under the scaled measure $\mu_{sc}$; see Appendix D.1 for details.

products between the frame elements and their derivatives. By replacing analytic derivations with a numerical frame operator formulation, SaFARi unifies previously distinct SSM variants under a single theoretical framework, enabling the construction of new models based on localized and adaptive bases.

**Scaled measure.** For the uniform scaled measure $\mu_t(\tau) = \frac{1}{t}\mathbf{1}_{[0,t]}(\tau)$ and warped frame

$$\varphi_n(\tau,t) = \varphi_n(\tau/t),$$

the SaFARi dynamics take the form

$$\dot{h}(t) = -\frac{1}{t}A_{sc}\,h(t) + \frac{1}{t}B_{sc}\,u(t), \qquad (6)$$

with

$$A_{sc} = I + U_\Upsilon U_{\tilde\varphi}^*, \qquad (B_{sc})_n = \varphi_n(1), \qquad (7)$$

where $\Upsilon\varphi_n(s) = s\,\partial_s\varphi_n(s)$ and $U_\varphi$ denotes the frame analysis operator. For Legendre polynomials, this construction recovers the HiPPO-LegS operator.

**Translated measure.** For the uniform translated measure $\mu_t(\tau) = \frac{1}{\theta}\mathbf{1}_{[t-\theta,t]}(\tau)$ and warped frame

$$\varphi_n(\tau,t) = \varphi_n\!\left(\frac{\tau-(t-\theta)}{\theta}\right),$$

the resulting dynamics are

$$\dot{h}(t) = -\frac{1}{\theta}A_{tr}\,h(t) + \frac{1}{\theta}B_{tr}\,u(t), \qquad (8)$$

with

$$A_{tr} = U_{\dot\varphi}U_{\tilde\varphi}^* + Q_\varphi Q_{\tilde\varphi}^*, \qquad (B_{tr})_n = \varphi_n(1), \quad (9)$$

where $\dot\varphi_n = \partial_s\varphi_n(s)$ and $Q_\varphi f = f(0)\varphi(0)$ is the zero-time frame operator. For Legendre polynomials, this reduces to the LMU (Voelker et al., 2019) and HiPPO-LegT updates.

## 3. Methodology: State Space Models over Wavelet Frames

In this work, we utilize the introduced SaFARi framework (Babaei et al., 2025) to construct our wavelet-based SSMs. As described, SaFARi generalizes the HiPPO formulation by providing a numerical method to derive state transition matrices for any frame, not just limited to an orthogonal basis. Leveraging this generality, we instantiate a collection of continuous and discrete wavelet frames to construct the dynamics of an SSM, yielding a novel multiscale state-space model we called *WaveSSM*.

In this section, we construct a set of time-localized wavelets of interest. Our objective is to design SSMs whose latent state evolves in the coefficient space induced by a wavelet frame $\Phi = \{\varphi_i\}_{i=0}^{N-1}$, whose atoms are well-localized in the underlying time–frequency tiling. The resulting *WaveSSMs* inherit their inductive bias directly from the chosen frames. In our experiments, we instantiate frames using continuous-wavelet constructions (Morlet, Gaussian-derivative, and Mexican hat), discrete wavelet transforms (as Daubechies), and Slepian-based frequency frames, which offer complementary control over spectral coverage.

### 3.1. Wavelet frame construction

Let $\Phi$ be a collection of $N$ time-localized atoms discretized on a grid of length $L$, forming a frame matrix $F \in \mathbb{R}^{N \times L}$ whose rows are sampled atoms. We consider real-valued multiscale constructions obtained by shifting and dilating a prototype waveform:

$$\varphi_{k,m}(t) \propto \psi\!\left(\frac{t-\tau_m}{\sigma_k}\right), \qquad (10)$$

where $\sigma_k$ controls temporal scale (width) and $\tau_m$ controls the shift. Each atom is subsequently normalized to unit

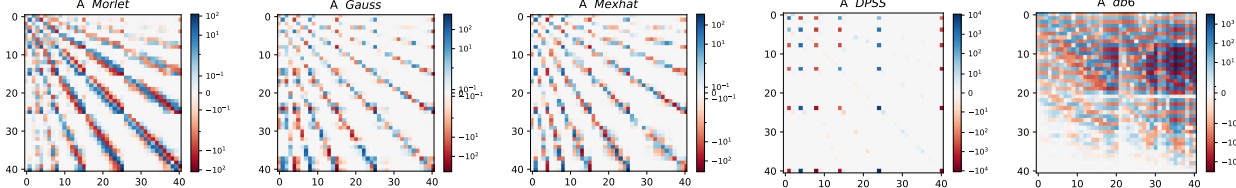

*Figure 3.* Visualization of the transition matrices $A$ obtained for the wavelet families for $N = 40$. From left to right: Morlet, Gaussian-derivative (Gauss), Mexican hat (Mexhat), Discrete Prolate Spheroidal Slepians (DPSS), and Daubechies (db6).

energy, $\|\varphi_{k,m}\|_2 = 1$. Complete theoretical definitions can be found in Appendix A.3 and practical implementation details on the frame construction are given in Appendix E. As a summary:

**Morlet wavelets.** Morlet frames are constructed from a real sinusoid modulated by a Gaussian envelope. Morlet frames are well-suited for representing localized, band-limited transients.

**Gaussian-derivative wavelets.** Gaussian-derivative atoms are obtained from the $P$-th derivative of a Gaussian envelope. In our experiments, we fix $P = 1$. Gaussian wavelets are particularly effective for capturing broadband, pulse-like transients and sharp events.

**Mexican hat.** The Mexican hat wavelet corresponds to the second derivative of a Gaussian. Increasing $P = 2$ introduces additional oscillations while preserving strong time localization. Additionally, their symmetric structure yields localized representations with minimal interference.

**DPSS-based frames.** Discrete prolate spheroidal sequences (DPSS), also known as Slepians, provide an alternative construction optimized for spectral concentration.

**Daubechies frames.** Daubechies atoms are obtained from a *discrete* orthonormal wavelet family with compact support and a specified number of vanishing moments. In our experiments, we use db6 wavelets.

### 3.2. Stability considerations

Our construction relies on the existence of a numerically stable dual frame $\widetilde{\Phi} = S^{-1}\Phi$, where $S = U_\Phi^* U_\Phi$ denotes the frame operator. In the continuous setting, stability corresponds to $S$ being invertible with bounded inverse, i.e., $\|S^{-1}\| < \infty$, ensuring that projections onto the dual do not amplify noise or discretization errors.

In practice, the frame $\Phi$ is implemented through a discretized matrix $F \in \mathbb{R}^{N \times L}$ obtained by sampling the $N$ frame atoms on a temporal grid of length $L$. With this convention, the relevant operator for coefficient-space projections is $S := FF^* \in \mathbb{R}^{N \times N}$. Because the wavelet frames considered in this work are generally redundant and

non-orthogonal, the conditioning of $S$ cannot be taken for granted and plays a central role in the stability of the induced state-space dynamics.

**Dual stability and its impact on the state-space dynamics.** As described in (7) and (9), the continuous-time state transition is obtained by projecting time derivatives of frame atoms back onto the span of the frame using the dual. In discrete form, this projection is realized via the least-squares problem

$$A \in \arg \min_{X \in \mathbb{R}^{N \times N}} \|\dot{F} - XF\|_F^2, \tag{11}$$

where $\dot{F} \in \mathbb{R}^{N \times L}$ stacks the row-wise time derivatives of the sampled atoms.

**Lemma 3.1.** *Let $F \in \mathbb{R}^{N \times L}$ be a discretized frame matrix with full row rank and let $S := FF^*$ denote the associated frame operator. Let $\dot{F}$ denote the row-wise time derivatives of the sampled frame atoms, and define $A$ by the above least-squares projection. Then $A$ is uniquely defined and satisfies*

$$A = \dot{F}F^* S^{-1}, \qquad \|A\|_2 \leq \frac{\|\dot{F}F^*\|_2}{\lambda_{\min}(S)}. \tag{12}$$

This shows that poor conditioning of S amplifies the magnitude of the transition matrix. Leading to large off-diagonal entries in $A$ and inducing non-local mixing between state components, which degrades truncation stability (Babaei et al., 2025) and can render the state dynamics numerically unstable.

**Frame tightening.** To improve numerical stability, we explicitly construct near-tight discretized frames by whitening the row space:

$$F \leftarrow S^{-1/2}F. \tag{13}$$

This transformation enforces $FF^* \approx I_N$, yielding a well-conditioned frame. Frame tightening eliminates near-linear dependencies between frame atoms, stabilizes the dual frame, and suppresses spurious amplification in derivative projections. See the experimental improvement in Fig 4.

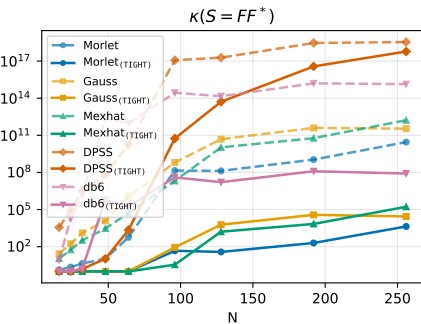

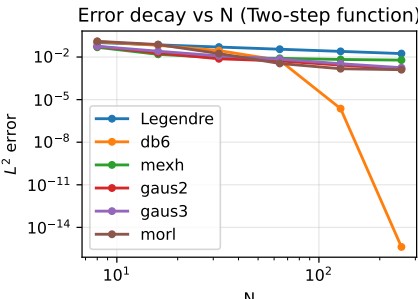

*Figure 4.* Condition number of the discretized frame operator $\kappa(S)$ as a function of the state dimension $N$. Raw frames (dashed) can become ill-conditioned as $N$ grows, while *tightening* $F$ (in solid lines) mitigates this degradation. The improvement is especially pronounced for Morlet, Gaussian, Daubechies, and Mexican-hat frames.

*Figure 5.* Approximation errors on various frames for the two-step piecewise constant function, see Appendix C for details.

### 3.3. Why wavelet frames outperform polynomials on localized irregularities

One of the starting motivational points of this paper is a standard approximation-theoretic fact: *localized multiscale wavelets represent localized irregularities (jumps, sharp transitions) much more efficiently than global polynomial modes*. To give a more precise but informal coating to this fact, we recall that given a dictionary $\mathcal{D} \subset L^2(0,1)$, its best $N$-term approximation error is defined as

$$\sigma_N^{\mathcal{D}}(f)_2 := \inf_{\substack{g \in \mathrm{span}(\mathcal{D}) \\ g \text{ uses } \leq N \text{ atoms}}} \|f - g\|_{L^2(0,1)}.$$

**Theorem 3.2** (Informal)**.** If $f$ is smooth on an interval $(0,1)$ except for $O(1) > 0$ breakpoints, then a wavelet-type Parseval frame $\Phi$ achieves

$$\sigma_N^{\Phi}(f)_2 \lesssim N^{-s} \qquad (f \in \mathrm{PW}H^s, \ s > \tfrac{1}{2}),$$

whereas even the best nonlinear $N$-term approximation using global Legendre/HiPPO-type polynomial modes has worst-case rate at best

$$\sigma_N^{\mathcal{L}}(f)_2 \gtrsim N^{-1/2}$$

*on the same class (the difference can already be seen on a simple class of functions with one jump discontinuity).*

In short: *wavelets exploit locality (few coefficients per scale near each breakpoint), while global polynomials cannot localize a jump*, which creates the rate gap $N^{-s}$ vs. $N^{-1/2}$.

The precise assumptions (multiscale localization, bounded overlap, and vanishing-moment/cancellation on smooth pieces) and complete proofs are given in Appendices A and B. Compact support is used there mainly to keep the arguments simple; analogous statements hold for non-compactly

supported but rapidly decaying wavelet systems, at the cost of more technical localization estimates.

We refer to Fig. 5 for a visual representation of the error decays for various wavelet frames and Legendre frames when approximating a two-step constant function using the best $N$-term approximations. We refer to the Appendix C for details on the experimental setting.

## 4. Why Linear Time-Invariant HiPPO Kernels Cannot Preserve Disjoint Temporal Windows

A natural question arises when comparing WaveSSM to polynomial-based state-space models: If HiPPO can, in principle, "store the entire input history" in its final state, would it be capable of storing and recalling *multiple disjoint windows* on demand? In this section, we clarify this distinction and show that SSMs based on HiPPO fundamentally lack this ability due to the superpositional nature of their convolutional kernels.

### 4.1. HiPPO as a Single Convolutional Memory Trace

An HiPPO layer with kernel $K \in \mathbb{R}^T$ computes its final state as a linear time-invariant convolution:

$$h_T = \sum_{\tau=1}^{T} K[\tau] \, x_{T-\tau} \qquad (14)$$

for an input sequence $(x_1, \ldots, x_T)$. Thus, the model forms a *single linear combination* of all past inputs, with each time index weighted by the same kernel shape $K$. This representation indeed contains global information about the input trajectory, but only in a compressed, linearly mixed form. Crucially, the LTI structure imposes that *all* time positions are projected through the same kernel and accumulated into the same state vector.

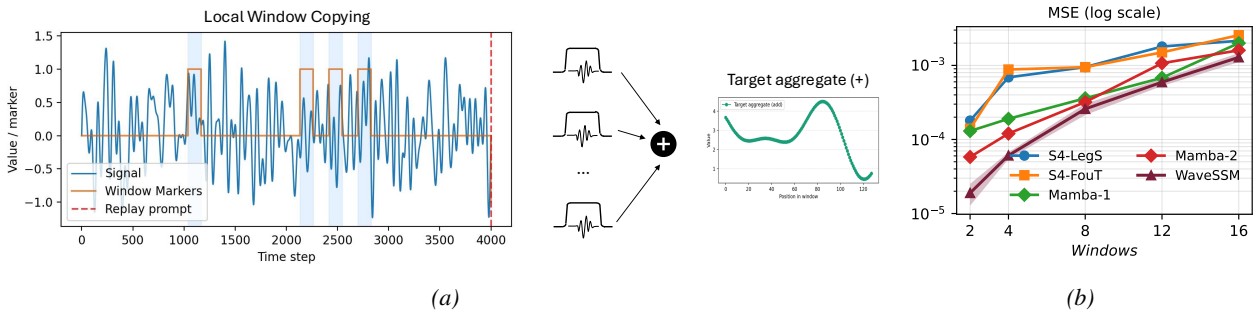

*(a)*            *(b)*

*Figure 6.* Continuous Window Copying Task. **Left:** A white-noise signal of length-$L$=4000 is streamed to the model together with sparse random markers defining non-overlapping windows of interest $W \in [2, 20]$. After a variable delay, the model must selectively retrieve the marked segments and apply a prescribed operation (i.e., addition) to generate the output. **Right:** Results reported as reconstruction MSE in log scale as a function of the number of windows $W$. For WaveSSM, we report the mean with confidence intervals across the five wavelet variants introduced in this work under the scaled measure.

## 4.2. Superposition Prevents Temporal Attention

Suppose two disjoint windows of the input occur at times $t_A$ and $t_B$, with values $w_A = \{x_t : t \in A\}$ and $w_B = \{x_t : t \in B\}$. From (14), their contributions to the final state are:

$$h_T = \sum_{t \in A} K[T-t]\, x_t + \sum_{t \in B} K[T-t]\, x_t + \text{(other times)} \tag{15}$$

Both windows are therefore *superimposed* through the same kernel weights. Because the mapping from each window to the final state shares the same functional form, the contributions of windows $A$ and $B$ are inseparable in general, i.e., the model does not possess distinct subspaces dedicated to different regions of time. Recovering $w_A$ without also recovering a mixture with $w_B$ is typically hard unless the windows have a special structure (e.g., are orthogonal under the kernel).

## 4.3. Storing the Whole History Is Not Storing It Usefully

The ability of HiPPO to preserve the entire history in its final state does not imply that this memory is *addressable*. Given only $h_T$, a decoder cannot recover $w_A$ and $w_B$ independently because they are linearly entangled through the shared kernel. In contrast, WaveSSM constructs its state representation using temporally localized frame elements. Each state dimension has a localized temporal receptive field, enabling the model to assign disjoint windows to disjoint subsets of coordinates with minimal interference. This yields an *attention-like* mechanism: multiple windows can be stored, preserved, and retrieved independently by design.

**Continuous Window Copying Task.**

We introduce this task to evaluate a model's ability to store and manipulate *temporally localized* information in an *addressable* manner (Fig. 6a). The input is a time series of length $T$, where the first channel is a real-valued signal $x(t)$ and the second channel contains sparse marker events in-

dicating the start and end of multiple windows of interest. The locations and durations of the windows are sampled randomly and do not overlap. At the end of the sequence, the model is instructed to apply a specified mathematical operation (e.g., *addition*) to the windows and return the results as an output vector.

We compare WaveSSM against both linear time-invariant SSM baselines, such as S4, and selective SSM baselines, namely Mamba-1 and Mamba-2. The inclusion of Mamba is important because its input-dependent dynamics provide a learned selection mechanism, and therefore test whether selectivity alone is sufficient for this form of addressable temporal storage. For a controlled architectural comparison, all models use the same two-layer backbone and model width $d_{\text{model}} = 128$. For Mamba, we keep the state sizes used by the standard configurations: $d_{\text{state}} = 16$ for Mamba-1 and $d_{\text{state}} = 128$ for Mamba-2. We do not force Mamba-1 to use $d_{\text{state}} = 128$, since this substantially increases the cost of the selective scan and departs from the recommended default configuration.

As shown in Fig. 6b, WaveSSM achieves roughly an order-of-magnitude lower reconstruction error than S4 under the same two-layer, 128-dimensional model-width setting. Mamba-1 and Mamba-2 improve over the linear S4 baselines, confirming that input-dependent selection is useful for the task. However, both remain consistently above WaveSSM in reconstruction error. This suggests that learned selectivity by itself does not fully solve the interference problem induced by storing several disjoint windows in a shared state. In contrast, WaveSSM's latent state is organized by localized multiscale frame elements; different marked windows activate largely distinct subsets of state coordinates, allowing multiple windows to be stored concurrently and read out with less cross-window interference.

*Table 1.* AUROC (↑) on the PTB-XL dataset for ECG multi-label/multi-class classification. The three best performing methods are highlighted in red[1] (First), blue[2] (Second), and violet[3] (Third). Result reproduced from (Karami et al., 2025).

| Model | Diag | Sub-Diag | Super-Diag | Form | Rhythm | Overall |
|---|---|---|---|---|---|---|
| Transformer (Vaswani et al., 2017b) | 0.876 | 0.882 | 0.887 | 0.771 | 0.831 | 0.849 |
| Spacetime (Zhang et al., 2023) | 0.941 | 0.933 | 0.929 | 0.883 | 0.967 | 0.931 |
| S4 (Gu et al., 2022b) | 0.939 | 0.929 | 0.931[3] | 0.895 | 0.977[2] | 0.934 |
| InceptionTime (Fawaz et al., 2020) | 0.931 | 0.930 | 0.921 | 0.899 | 0.953 | 0.927 |
| Wavelet features (Strodthoff et al., 2021) | 0.855 | 0.859 | 0.874 | 0.757 | 0.890 | 0.847 |
| Mamba (Gu & Dao, 2023) | 0.929 | 0.905 | 0.912 | 0.876 | 0.952 | 0.915 |
| MS-SSM (Karami et al., 2025) | 0.939 | 0.935 | 0.930 | 0.899 | 0.980[1] | 0.937 |
| WaveSSM$_{MorletS}$ | 0.943 | 0.938 | 0.931[3] | 0.906 | 0.974[3] | 0.938 |
| WaveSSM$_{GaussS}$ | 0.945[3] | 0.936 | 0.930 | 0.904 | 0.974[3] | 0.938 |
| WaveSSM$_{MexhatS}$ | 0.946[2] | 0.939 | 0.931[3] | 0.914[1] | 0.970 | 0.940[3] |
| WaveSSM$_{dpssS}$ | 0.941 | 0.940[3] | 0.931[3] | 0.913[2] | 0.969 | 0.939 |
| WaveSSM$_{db6S}$ | 0.945[3] | 0.941[2] | 0.932[2] | 0.914[1] | 0.972 | 0.941[2] |
| WaveSSM$_{MorletT}$ | 0.945[3] | 0.940[3] | 0.930 | 0.902 | 0.968 | 0.937 |
| WaveSSM$_{GaussT}$ | 0.938 | 0.932 | 0.926 | 0.902 | 0.973 | 0.934 |
| WaveSSM$_{MexhatT}$ | 0.944 | 0.940[3] | 0.932[2] | 0.909[3] | 0.970 | 0.939 |
| WaveSSM$_{dpssT}$ | 0.947[1] | 0.937 | 0.931[3] | 0.906 | 0.967 | 0.938 |
| WaveSSM$_{db6T}$ | 0.946[2] | 0.943[1] | 0.934[1] | 0.913[2] | 0.973 | 0.942[1] |

*Table 2.* Univariate long sequence time-series forecasting results on Informer dataset reported as MSE (↓). Highlighted in red[1] (First), blue[2] (Second), and violet[3] (Third). Full results in Appendix G.

| Model | ETTh$_1$ | ETTm$_1$ | Weather | ECL |
|---|---|---|---|---|
| S4 | 0.116[3] | 0.292 | 0.245 | 0.432 |
| Informer | 0.269 | 0.512 | 0.359 | 0.582 |
| LogTrans | 0.273 | 0.598 | 0.388 | 0.624 |
| Reformer | 2.112 | 1.793 | 2.087 | 7.019 |
| Prophet | 2.735 | 2.747 | 3.859 | 6.901 |
| WaveSSM$_{MorletS}$ | 0.114[2] | 0.261 | 0.241 | 0.283 |
| WaveSSM$_{GaussS}$ | 0.144 | 0.245[2] | 0.218[1] | 0.273[2] |
| WaveSSM$_{MexhatS}$ | 0.128 | 0.247 | 0.233 | 0.283 |
| WaveSSM$_{dpssS}$ | 0.160 | 0.260 | 0.232 | 0.268[1] |
| WaveSSM$_{db6S}$ | 0.117 | 0.264 | 0.225 | 0.301 |
| WaveSSM$_{MorletT}$ | 0.135 | 0.293 | 0.233 | 0.316 |
| WaveSSM$_{GaussT}$ | 0.102[1] | 0.256 | 0.232 | 0.312 |
| WaveSSM$_{MexhatT}$ | 0.130 | 0.245[2] | 0.220[2] | 0.310 |
| WaveSSM$_{dpssT}$ | 0.138 | 0.231[1] | 0.223[3] | 0.277[3] |
| WaveSSM$_{db6T}$ | 0.122 | 0.246[3] | 0.229 | 0.313 |

*Table 3.* **SC35 classification.** Classification accuracy (↑) in unprocessed audio signals ($L = 16,000$) and test-time frequency shift at half the training frequency. Highlighted in red[1] (First), blue[2] (Second), and violet[3] (Third). All results are reproduced in this work for 3 different seeds.

| Model | Raw(16kHz) | 0.5×(8kHz) |
|---|---|---|
| S4$_{LegS}$ | 96.08 $\pm0.15$ | 91.32 $\pm0.17$[2] |
| S4$_{FouT}$ | 95.27 $\pm0.20$ | 91.59 $\pm0.23$[1] |
| WaveSSM$_{MorletS}$ | 96.42 $\pm0.26$[3] | 90.14$\pm0.20$ |
| WaveSSM$_{GaussS}$ | 96.47 $\pm0.16$[2] | 90.84$\pm0.13$[3] |
| WaveSSM$_{MexhatS}$ | 96.17 $\pm0.16$ | 89.70 $\pm0.31$ |
| WaveSSM$_{dpssS}$ | 96.09 $\pm0.28$ | 89.61 $\pm0.15$ |
| WaveSSM$_{MorletT}$ | 96.27 $\pm0.13$ | 86.20 $\pm0.25$ |
| WaveSSM$_{GaussT}$ | 96.17 $\pm0.21$ | 89.81 $\pm0.31$ |
| WaveSSM$_{MexhatT}$ | 96.27 $\pm0.14$ | 84.14 $\pm0.22$ |
| WaveSSM$_{dpssT}$ | 96.55 $\pm0.22$[1] | 90.78 $\pm0.27$ |

substantially improves numerical stability compared to unconstrained dense matrices, whose non-normality can lead to ill-conditioned powers and unstable kernel computation.

## 5. Incorporating Wavelet Frames into the S4 Architecture

While wavelet frames induce a dense continuous-time state transition operator, we do not parameterize or learn the full state matrix $A \in \mathbb{R}^{N \times N}$. Instead, we embed the wavelet-initialized dynamics into the S4 architecture (Gu et al., 2022b), using its diagonal-plus-low-rank (DPLR) parameterization. This choice provides several advantages. DPLR reduces the parameter count and per-step computation from $O(N^2)$ to $O(N)$; furthermore, it enables efficient computation. Also, constraining $A$ to a normal-plus-low-rank form

## 6. Experimentation

Having shown that wavelet frames yield an *addressable*, time-local state representation and that these wavelet-initialized dynamics can be realized efficiently within the S4 DPLR parameterization, we now test whether this inductive bias translates into measurable gains on real sequence modeling problems. In particular, we focus on settings where non-stationary or transient structure is central (e.g., biosignals), while also reporting on more challenging long-context benchmarks as the Long Range Arena.

*Table 4.* Accuracy (↑) on the Long Range Arena (LRA) benchmark. "✗" indicates infeasible to learn a non-trivial solution. Highlighted in red[1] (First), blue[2] (Second), and violet[3] (Third). Extended comparison can be found in Table S8 and hyperparameter settings are detailed in Table S5.

| Method | ListOps (2048) | Text (4096) | Retrieval (4000) | Image (1024) | Pathfinder (1024) | PathX (16,384) |
|---|---|---|---|---|---|---|
| Transformer | 36.37 | 64.27 | 57.46 | 42.44 | 71.40 | ✗ |
| $S4_{LegS}$ | 59.60 | 86.82 | 90.90 | 88.65 | 94.20 | 96.35[1] |
| $S4_{FouT}$ | 57.88 | 86.34 | 89.66 | 89.07 | 94.46[2] | ✗ |
| $WaveSSM_{MorletS}$ | 60.88 | 89.11[3] | 91.97[1] | 89.26 | 94.94[1] | ✗ |
| $WaveSSM_{GaussS}$ | 59.62 | 88.06 | 91.19 | 89.68[3] | ✗ | ✗ |
| $WaveSSM_{MexhatS}$ | 60.13 | 89.33[1] | 91.88[2] | 89.35 | ✗ | ✗ |
| $WaveSSM_{dpssS}$ | 61.74[2] | 88.56 | 91.52 | 89.41 | ✗ | ✗ |
| $WaveSSM_{MorletT}$ | 61.79[1] | 89.23[2] | 91.88[2] | 89.86[1] | 73.31 | ✗ |
| $WaveSSM_{GaussT}$ | 60.93 | 87.65 | 90.89 | 87.26 | 94.33[3] | ✗ |
| $WaveSSM_{MexhatT}$ | 61.34[3] | 89.05 | 91.68[3] | 89.74[2] | 73.42 | ✗ |
| $WaveSSM_{dpssT}$ | 60.98 | 89.01 | 91.48 | 89.04 | 62.51 | ✗ |

*Remark* 6.1. A natural concern is that replacing globally supported bases with highly localized wavelet frames would require a substantially larger state dimension to enough maintain coverage along the input sequence. In our experiments, *this overhead does not arise*. During the experimentation, we use the *same* order as polynomial-based baselines, to perform a fair comparison. Details on the hyperparameters could be found in Table S5.

### 6.1. ECG Transient Morphology

Electrocardiograms (ECGs) are commonly used as one of the first examination tools for assessing and diagnosing cardiovascular diseases. However, its interpretation remains a challenging task. We use the publicly available PTB-XL dataset (Wagner et al., 2020), consisting of recordings of 10 seconds each, annotated with 71 diagnostic statements following the SCP-ECG standard. In this setting, WaveSSM is particularly well matched as many clinically relevant ECG findings, such as arrhythmias, conduction blocks, or ischemic patterns, are characterized by localized and transient waveform morphology. Table 1 reports AU-ROC results across all PTB-XL classification tasks, where WaveSSM variants consistently achieve the strongest overall performance across nearly all categories. In particular, the Daubiches-derived models perform particularly well, with $WaveSSM_{db6T}$ attaining the highest overall score. The most pronounced gains appear in the Form and Diag tasks; only on the Rhythm task do state-of-the-art models such as MS-SSM (Karami et al., 2025) remain marginally superior.

### 6.2. Time Series Forecasting

We also evaluate our model on the standard Informer long-sequence forecasting benchmarks (Zhou et al., 2021), which include the Electricity Transformer Temperature (ETT), Weather, and Electricity Consumption Load (ECL) benchmarks. We follow the experimental protocol of (Gu et al.,

2022b). Table 2 reports univariate forecasting results measured by mean squared error (MSE) for the longest prediction horizon evaluated on each task (details on the horizons are provided in Appendix F.2, with extended results for both univariate and multivariate predictions in Appendix G). Across all datasets, WaveSSM variants achieve consistently strong performance. Notably, WaveSSM improves upon its polynomial-based counterpart, S4, on most of the evaluated tasks, with the exception of a few time horizons.

### 6.3. Raw Speech Classification

The Speech Commands (SC35) (Warden, 2018) is a 35-way spoken-word classification task using raw audio. Each input is a 1-second signal sampled at 16 kHz. While previous studies typically extract spectral features, state-space models like S4 have demonstrated the ability to learn directly from time-domain inputs. As shown in Table 3, in the in-distribution setting (Raw, 16 kHz), WaveSSM surpasses S4 baselines, achieving accuracies around 96.5%. We also quantify the performance gains under a zero-shot resampling scenario, where each test waveform is uniformly subsampled to 8 kHz without retraining; however, under this frequency shift, S4 still remains more robust.

### 6.4. Long Range Arena

Lastly, we evaluate in the more general Long Range Arena (LRA) benchmark (Tay et al., 2021), which is a widely used suite of challenging long-context tasks designed to evaluate a model's ability to capture long-range dependencies across diverse modalities, including symbolic reasoning over long texts and serialized images.

As shown in Table 4, WaveSSM variants achieve consistently strong results across LRA tasks when directly compared to S4. In particular, WaveSSM surpasses it on four out of six tasks: ListOps, Text, Retrieval, and Image classifica-

tion. However, WaveSSM exhibits limitations in very long-context regimes: some variants do not scale to Pathfinder, and all WaveSSM variants are infeasible on PathX (similarly to most of S4 variants). We view this as a limitation of the current formulation, suggesting a trade-off between the strong locality and multiscale inductive bias that benefits transient signals and the global, very long-range dependencies required by the hardest LRA settings (Solozabal et al., 2025; Amos et al., 2024).

## 7. Conclusions & Limitations

WaveSSM shows that replacing globally supported orthogonal bases with localized wavelet frames yields an interesting state representation for sequence modeling. By inducing continuous-time SSM dynamics directly from frame projections, the hidden state becomes temporally addressable: disjoint regions of the input activate largely separate subsets of state, enabling selective retrieval and manipulation of multiple non-overlapping windows, achieving an *attention-like* behavior.

## Impact Statement

This paper presents work whose goal is to advance the field of Machine Learning. There are many potential societal consequences of our work, none of which we feel must be specifically highlighted here.

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

# *Supplementary Material:*
# **Multiscale State-Space Models for Non-stationary Signal Attention**

## A. Frames and wavelets

We briefly present the framework that we worked on for this paper. The details can be found in any treatise of frames and wavelets, such as (Christensen et al., 2003).

### A.1. Frames

We work in the Hilbert space $L^2(0,1)$ with inner product $\langle f, g \rangle := \int_0^1 f(t)g(t)\,dt$ and norm $\|f\|_2 := \langle f, f \rangle^{1/2}$.

**Frames.**  A countable family $\Phi = \{\phi_\lambda\}_{\lambda \in \Lambda} \subset L^2(0,1)$ is a *frame* if there exist constants $0 < A \le B < \infty$ such that for all $f \in L^2(0,1)$,

$$A\|f\|_2^2 \ \le \ \sum_{\lambda \in \Lambda} |\langle f, \phi_\lambda \rangle|^2 \ \le \ B\|f\|_2^2. \tag{16}$$

We say that the frame is *tight* if $A = B$, and *Parseval* if $A = B = 1$.

Let $U : L^2(0,1) \to \ell^2(\Lambda)$ be the *analysis operator*, defined component-wise as $(Uf)_\lambda := \langle f, \phi_\lambda \rangle$. Then the corresponding *frame operator* is given by $S := U^*U$, or explicitly as

$$Sf = \sum_{\lambda \in \Lambda} \langle f, \phi_\lambda \rangle \, \phi_\lambda.$$

Note that for a Parseval frame, $S = I$ and $U$ is an isometry $\|Uf\|_{\ell^2} = \|f\|_2$, while for a tight frame we have a similar reconstruction expression $f = \frac{1}{A} \sum_{\lambda \in \Lambda} \langle f, \phi_\lambda \rangle \phi_\lambda$.

**Nonlinear $N$-term approximation in a dictionary/frame.**  For a countable dictionary $\mathcal{D} = \{\varphi_\lambda\}_{\lambda \in \Lambda}$ we define

$$\sigma_N^{\mathcal{D}}(f)_2 := \inf_{\substack{\Gamma \subset \Lambda \\ |\Gamma| = N}} \ \inf_{\{c_\lambda\}_{\lambda \in \Gamma}} \left\| f - \sum_{\lambda \in \Gamma} c_\lambda \varphi_\lambda \right\|_2. \tag{17}$$

We deal with nonlinear $N$-approximations in the Appendix B.

**A key truncation inequality for Parseval frames.**  The following lemma will be used in the proof of Theorem B.1. In particular, it replaces the "Parseval tail identity" used for orthonormal bases.

**Lemma A.1** (Truncation bound for Parseval frames). *Assume $\Phi = \{\phi_\lambda\}_{\lambda \in \Lambda}$ is a Parseval frame. For any subset $\Gamma \subset \Lambda$, define the truncated synthesis ("keep coefficients in $\Gamma$")*

$$T_\Gamma f \ := \ \sum_{\lambda \in \Gamma} \langle f, \phi_\lambda \rangle \, \phi_\lambda.$$

*Then for all $f \in L^2(0,1)$,*

$$\|f - T_\Gamma f\|_2^2 \ \le \ \sum_{\lambda \notin \Gamma} |\langle f, \phi_\lambda \rangle|^2. \tag{18}$$

*Proof.* Let $U$ be the analysis operator. For a Parseval frame, $S = U^*U = I$. Let $P_\Gamma : \ell^2(\Lambda) \to \ell^2(\Lambda)$ be the coordinate projection that zeroes out indices not in $\Gamma$. Then $T_\Gamma = U^* P_\Gamma U$ and

$$f - T_\Gamma f = U^*(I - P_\Gamma)Uf.$$

Using $\|U^*\| = 1$ for Parseval frames (since $U$ is an isometry), we obtain

$$\|f - T_\Gamma f\|_2 \le \|(I - P_\Gamma)Uf\|_{\ell^2}.$$

Finally, squaring the last expression gives us (18) because $\|(I - P_\Gamma)Uf\|_{\ell^2}^2 = \sum_{\lambda \notin \Gamma} |\langle f, \phi_\lambda \rangle|^2$.  $\square$

## A.2. Wavelets

We briefly recall wavelet constructions through the standard operations of scaling, translation and modulation. A classical reference for a more detailed (and more general) treatment is (Daubechies, 1992).

**Mother wavelet.**    A function $\psi : \mathbb{R} \to \mathbb{R}$ is called *mother wavelet* if the the following conditions hold:

1. $\int_{\mathbb{R}} |\psi(t)| dt < \infty$,

2. $\int_{\mathbb{R}} |\psi(t)|^2 dt < \infty$,

although sometimes in the practice one substitutes the first condition with $\int_{\mathbb{R}} \psi(t) dt = 0$, i.e. it is required that the function $\psi$ has zero mean and finite energy (it is in $L^2(\mathbb{R})$).

**Scaling and translation.**    Given a mother wavelet $\psi$, one constructs a family of wavelets of the form

$$\psi_{a,b}(t) := \frac{1}{\sqrt{a}} \psi\left(\frac{t-b}{a}\right), \quad a \neq 0, b \in \mathbb{R}.$$

The intuition behind this is that the translation allows us to "focus" the wavelet to the desired locality in $\mathbb{R}$, while scaling allows us to reduce the "window of focus" as desired.

**Vanishing moments.**    A (real) mother wavelet $\psi$ on $\mathbb{R}$ is said to have $p$ vanishing moments if

$$\int_{\mathbb{R}} t^m \psi(t)\, dt = 0, \qquad m = 0, 1, \ldots, p-1. \tag{19}$$

This is the analytic mechanism behind small coefficients for smooth functions, a property we will assume in Theorem B.1, see (Daubechies, 1992).

**Modulation.**    In some constructions, one also considers *modulated wavelets*, obtained by multiplying the mother wavelet by a complex exponential (or its trigonometric components). Given a real-valued mother wavelet $\psi$ and a frequency parameter $\xi \in \mathbb{R}$, one defines

$$\psi^{(\xi)}(t) := e^{i\xi t} \psi(t),$$

and correspondingly

$$\psi_{a,b}^{(\xi)}(t) = \frac{1}{\sqrt{a}} e^{i\xi t} \psi\left(\frac{t-b}{a}\right).$$

While scaling primarily controls the frequency range of the wavelet (with smaller scales corresponding to higher frequencies), modulation allows one to shift the frequency content within a given scale. This leads to a finer localization in the time–frequency plane and is particularly useful in applications where oscillatory behavior at comparable scales but different frequencies needs to be resolved. Such constructions appear, for example, in wavelet packet decompositions and in connections between wavelet analysis and time–frequency methods (Mallat, 1999).

## A.3. The wavelet frames used in WaveSSM

In the main paper, the latent state is defined in the coefficient space of a *time–frequency frame* $\Phi = \{\phi_n\}_{n=0}^{N-1}$ sampled on a grid of length $L$, giving a discretized frame matrix $F \in \mathbb{R}^{N \times L}$ whose rows are sampled atoms. The general atom form is

$$\phi_{k,m,l}(t) = \psi_k\left(\frac{t - \tau_m}{\sigma_k}\right) g_\ell(t), \tag{20}$$

where $\sigma_k$ is a temporal scale, $\tau_m$ is a time shift, and $g_\ell$ is a modulation factor. In particular we construct wavelet family with scaling, translation and modulation involved.

**Morlet (real) atoms.** Morlet atoms are real sinusoids modulated by a Gaussian envelope:

$$\phi_{k,m,\ell}(t) \;=\; \exp\Big(-\frac{(t-\tau_m)^2}{2\sigma_k^2}\Big)\cos(\omega_\ell t). \tag{21}$$

They are strongly localized in time and frequency.

**Gaussian derivative atoms.** For integer order $P \geq 1$, Gaussian derivative atoms are successive derivatives of a Gaussian:

$$\phi_{k,m}(t) \;=\; \frac{d^P}{dt^P}\exp\Big(-\frac{(t-\tau_m)^2}{2\sigma_k^2}\Big). \tag{22}$$

These atoms have at least zero-mean (cancellation) for $P \geq 1$ due to being derivatives of a fast-decaying function.

**Mexican hat.** The Mexican hat wavelet is the second derivative of a Gaussian (a special case of (22) with $P = 2$), commonly used for broadband, pulse-like transients.

**DPSS atoms.** Discrete prolate spheroidal sequences (DPSS), also known as Slepians, provide an alternative construction optimized for spectral concentration. Given a window length $M$ and bandwidth $W$, DPSS tapers $v_k$ (Slepian, 1978) maximize energy concentration within the specified frequency band. To construct a DPSS frame over a signal of length $L$, we select a set of bandwidths $W_k$ and, for each compute a corresponding set of DPSS tapers. Each taper is translated to multiple center locations $\tau_{k,m}$ across the signal domain.

$$\phi_{k,m}(t) = v_k(t - \tau_{k,m}). \tag{23}$$

# B. Function approximation via Parseval wavelet frames

## B.1. Setting

We compare nonlinear $N$-term approximation in: (i) a (tight) wavelet frame $\Phi$ (localized atoms), and (ii) the global polynomial dictionary $\mathcal{L}$ (shifted Legendre atoms), both under the same $N$-term budget. Although the following result holds with general polynomial dictionaries, we present the result for Legendre family, as it is more relevant to the SSM setting. We refer to Appendix A.1 for the relevant definition of non-linear $N$-approximation, while for more general approximation results of this type, we refer to (DeVore, 1998; Cohen et al., 1993).

**Global polynomial dictionary (HiPPO/Legendre).** Let $\mathcal{L} = \{\ell_n\}_{n\geq 0}$ be the orthonormal shifted Legendre functions on $[0,1]$:

$$\ell_n(t) := \sqrt{2n+1}\, P_n(2t-1), \qquad n \geq 0,$$

where $P_n$ is the $n$-th (standard) Legendre polynomial on $[-1,1]$. Then we have $\langle \ell_n, \ell_m \rangle = \delta_{nm}$.

**Piecewise Sobolev class (finite change points).** To simplify the notation and the argument that follows, we restrain ourselves to Sobolev spaces and in particular, their following straight-forward generalization. Let $s > 0$ be fixed and let $M \in \mathbb{N}$. We say $f \in \mathrm{PW}H_M^s(0,1)$ if there exist breakpoints $0 = t_0 < t_1 < \cdots < t_M = 1$ such that each restriction $f|_{(t_{m-1}, t_m)} \in H^s(t_{m-1}, t_m)$, where $H^s(t_{m-1}, t_m)$ is Sobolev space of functions (see for example (Adams & Fournier, 2003)).

This setting clearly allows jump discontinuities at $\{t_1, \ldots, t_{M-1}\}$.

**Theorem B.1** (Parseval wavelet frames approximations). *Assume $\Phi = \{\phi_\lambda\}_{\lambda \in \Lambda}$ is a Parseval frame in $L^2(0,1)$ such that:*

*a) The functions (atoms) $\phi_\lambda$ are multiscale localized in time, i.e. there exists a scale index $j(\lambda) \in \mathbb{Z}$ such that*

$$\mathrm{diam}(\mathrm{supp}(\phi_\lambda)) \lesssim 2^{-j(\lambda)}, \qquad \#\{\lambda : j(\lambda) = j\} \asymp 2^j. \tag{24}$$

*b) Only $O(1)$ functions $\phi_\lambda$ overlap any point at a given scale, i.e. there exists a constant $C$ such that*

$$\sup_{x \in (0,1)} \#\{\lambda \mid j(\lambda) = j,\, x \in supp(\phi_\lambda)\} \leq C.$$

c) *Assume further that* $\Phi$ *has sufficient cancellation/smoothness so that the following* coefficient decay on smooth pieces *holds: for each $s > 0$ there is $C_s$ such that if $f \in H^s(a,b)$ and $\mathrm{supp}(\phi_\lambda) \subset (a,b)$ then*

$$|\langle f, \phi_\lambda \rangle| \ \leq \ C_s \, 2^{-j(\lambda)\,(s+1/2)} \, \|f\|_{H^s(a,b)}. \tag{25}$$

*(For standard wavelet-type systems this is the usual vanishing-moment/Taylor-cancellation estimate.)*

*In addition, assume that $\phi_\lambda$ satisfy the wavelet-type scaling bound $\|\phi_\lambda\|_{L^1(0,1)} \ \leq \ C_1 \, 2^{-j(\lambda)/2}$ for all $(\lambda \in \Lambda)$.*

*Then:*

1. **(Wavelet-frame rate on piecewise smooth signals)** *If $s > \frac{1}{2}$ and $f \in \mathrm{PW}H_M^s(0,1)$, then*

$$\sigma_N^\Phi(f)_2 \ \leq \ C \, N^{-s}, \tag{26}$$

*where $C$ depends on $s$, $M$ and the frame constants in (24)–(25), but not on $N$.*

2. **(Global Legendre modes cannot beat $N^{-1/2}$ on a jump, even nonlinearly)** *There exists $f_\star \in \mathrm{PW}H_2^s(0,1)$ (a step function) and $c > 0$ such that for all $N \geq 1$,*

$$\sigma_N^{\mathcal{L}}(f_\star)_2 \ \geq \ c \, N^{-1/2}. \tag{27}$$

*Proof.* **Proof of 1.** Let $f \in \mathrm{PW}H_M^s(0,1)$ with breakpoints $0 = t_0 < t_1 < \cdots < t_{M-1} < t_M = 1$. For each scale index $j \in \mathbb{Z}$, let us define

$$\Lambda_j := \{\lambda \in \Lambda : \ j(\lambda) = j\}, \qquad \Lambda_j^{\mathrm{reg}} := \{\lambda \in \Lambda_j : \ \mathrm{supp}(\phi_\lambda) \subset (t_{m-1}, t_m) \text{ for some } m\},$$

and $\Lambda_j^{\mathrm{sing}} := \Lambda_j \setminus \Lambda_j^{\mathrm{reg}}$. By the condition *b)* from the theorem, each breakpoint $t_m$ is contained in the supports of at most $C$ atoms from $\Lambda_j$, hence

$$\#\Lambda_j^{\mathrm{sing}} \ \leq \ C \, (M - 1) \qquad \text{for all } j. \tag{28}$$

Also, by (24), $\#\Lambda_j \asymp 2^j$.

We split the argument in two cases, covering regular coefficients coming from $\Lambda_j^{\mathrm{reg}}$ and singular coming from $\Lambda_j^{\mathrm{sing}}$.

*Regular coefficients.* For $\lambda \in \Lambda_j^{\mathrm{reg}}$ we can apply (25) on the interval $(t_{m-1}, t_m)$ containing $\mathrm{supp}(\phi_\lambda)$ to obtain:

$$|\langle f, \phi_\lambda \rangle| \ \leq \ C_s \, 2^{-j(s+\frac{1}{2})} \|f\|_{H^s(t_{m-1}, t_m)}.$$

Summing over $\Lambda_j^{\mathrm{reg}}$ and using $\#\Lambda_j^{\mathrm{reg}} \leq \#\Lambda_j \lesssim 2^j$ yields

$$\sum_{\lambda \in \Lambda_j^{\mathrm{reg}}} |\langle f, \phi_\lambda \rangle|^2 \ \lesssim \ 2^j \cdot 2^{-2j(s+\frac{1}{2})} \Big( \sum_{m=1}^M \|f\|_{H^s(t_{m-1}, t_m)}^2 \Big) \ \lesssim \ 2^{-2js} \|f\|_{\mathrm{PW}H_M^s}^2, \tag{29}$$

where $\|f\|_{\mathrm{PW}H_M^s}^2 := \sum_{m=1}^M \|f\|_{H^s(t_{m-1}, t_m)}^2$.

*Singular coefficients.* We next use the scaling bound

$$\|\phi_\lambda\|_{L^1(0,1)} \ \leq \ C_1 \, 2^{-j(\lambda)/2} \qquad (\lambda \in \Lambda). \tag{30}$$

Then for any $f \in L^\infty(0,1)$, Hölder inequality implies

$$|\langle f, \phi_\lambda \rangle| \ \leq \ \|f\|_{L^\infty} \|\phi_\lambda\|_{L^1} \ \lesssim \ \|f\|_{L^\infty} \, 2^{-j(\lambda)/2}. \tag{31}$$

Note that here we used that since $f \in \mathrm{PW}H_M^s$ and $s > \frac{1}{2}$, the Sobolev embeddings $H^s(I_m) \hookrightarrow L^\infty(I_m)$ for each segment $I_m = (t_{m-1}, t_m)$, together with (30) imply the above inequality.

Let $c_\lambda := \langle f, \phi_\lambda \rangle$ and let $(c_k^*)_{k \geq 1}$ be the nonincreasing rearrangement of $(|c_\lambda|)_{\lambda \in \Lambda}$. If we let

$$T_N f := \sum_{\lambda \in \Lambda_N} c_\lambda \, \phi_\lambda, \qquad \Lambda_N := \{N \text{ indices corresponding to the largest } |c_\lambda|\},$$

then since $\Phi$ is Parseval, the canonical best $N$-term approximation Lemma A.1 implies

$$\|f - T_N f\|_2^2 \leq \sum_{k > N} (c_k^*)^2. \tag{32}$$

Now consider only the singular coefficients $\{c_\lambda : \lambda \in \Lambda^{\text{sing}}\}$. By (28) there are at most $C(M-1)$ of them at each scale $j$, and by (31) each such coefficient is $\lesssim 2^{-j/2}$. Therefore, after selecting the $N$ largest coefficients overall, the singular coefficients that can remain unselected must come from sufficiently fine scales. More precisely, let

$$J := \left\lfloor \frac{N}{C(M-1)} \right\rfloor.$$

Then the total number of singular coefficients with scale $j \leq J$ is at most $C(M-1)J \leq N$, hence the top-$N$ set $\Lambda_N$ may be chosen to include the singular coefficients with $j \leq J$. Consequently,

$$\sum_{\substack{\lambda \in \Lambda^{\text{sing}} \\ \lambda \notin \Lambda_N}} |c_\lambda|^2 \leq \sum_{j > J} \sum_{\lambda \in \Lambda_j^{\text{sing}}} |c_\lambda|^2 \lesssim \sum_{j > J} \#\Lambda_j^{\text{sing}} \cdot 2^{-j} \|f\|_\infty^2 \lesssim \|f\|_\infty^2 \sum_{j > J} 2^{-j} \lesssim \|f\|_\infty^2 \, 2^{-J}. \tag{33}$$

In particular, the singular tail is *exponentially small in $N$*:

$$\sum_{\substack{\lambda \in \Lambda^{\text{sing}} \\ \lambda \notin \Lambda_N}} |c_\lambda|^2 \lesssim \|f\|_\infty^2 \, 2^{-cN/M} \leq C_{s,M} \|f\|_{\text{PW}H_M^s}^2 \, N^{-2s}, \tag{34}$$

where the last inequality uses that $2^{-cN/M} \leq C_{s,M} N^{-2s}$ for all $N \geq 1$ and for sufficiently large constant $C_{s,M}$.

*Putting regular and singular parts together.*

The regular coefficients satisfy the scale-wise square-sum bound (29), which implies that the decreasing rearrangement of the regular part obeys $c_k^* \lesssim k^{-(s+\frac{1}{2})} \|f\|_{\text{PW}H_M^s}$, hence

$$\sum_{k > N} (c_k^*)^2 \lesssim \|f\|_{\text{PW}H_M^s}^2 \sum_{k > N} k^{-(2s+1)} \lesssim \|f\|_{\text{PW}H_M^s}^2 \, N^{-2s}.$$

Adding the singular tail estimate (34) gives

$$\|f - T_N f\|_2^2 \lesssim \|f\|_{\text{PW}H_M^s}^2 \, N^{-2s}, \qquad \text{i.e.} \qquad \|f - T_N f\|_2 \lesssim \|f\|_{\text{PW}H_M^s} \, N^{-s}.$$

The inequality (26) follows.

**Proof of 2.**

Let

$$f_\star(t) := \mathbf{1}_{[0,1/2]}(t).$$

Then $f_\star \in \text{PW}H_2^s(0,1)$ for every $s \geq 0$, since $f_\star$ is constant on each of the two subintervals $(0, \frac{1}{2})$ and $(\frac{1}{2}, 1)$.

Let us define the coefficients

$$a_n := \langle f_\star, \ell_n \rangle = \int_0^{1/2} \ell_n(t) \, dt.$$

With the change of variables $x = 2t - 1$ (so $dt = \frac{1}{2} dx$ and $t \in [0, 1/2] \iff x \in [-1, 0]$), we get

$$a_n = \frac{\sqrt{2n+1}}{2} \int_{-1}^0 P_n(x) \, dx. \tag{35}$$

Use the classical antiderivative identity (valid for $n \geq 1$)

$$\int P_n(x)\, dx = \frac{P_{n+1}(x) - P_{n-1}(x)}{2n+1},$$ (36)

to evaluate the definite integral in (35):

$$\int_{-1}^{0} P_n(x)\, dx = \left. \frac{P_{n+1}(x) - P_{n-1}(x)}{2n+1} \right|_{x=-1}^{x=0} = \frac{P_{n+1}(0) - P_{n-1}(0)}{2n+1} - \frac{P_{n+1}(-1) - P_{n-1}(-1)}{2n+1}.$$

Since $P_m(-1) = (-1)^m$, the $x = -1$ term vanishes: $P_{n+1}(-1) - P_{n-1}(-1) = (-1)^{n+1} - (-1)^{n-1} = 0$. Hence

$$\int_{-1}^{0} P_n(x)\, dx = \frac{P_{n+1}(0) - P_{n-1}(0)}{2n+1}, \qquad \text{so} \qquad a_n = \frac{P_{n+1}(0) - P_{n-1}(0)}{2\sqrt{2n+1}}.$$ (37)

*Even coefficients vanish.* If $n = 2m$ is even, then $n \pm 1$ are odd and $P_{\mathrm{odd}}(0) = 0$, so (37) gives $a_{2m} = 0$.

*Odd coefficients are $\gtrsim 1/n$.* If $n = 2m + 1$ is odd, then $n - 1 = 2m$ and $n + 1 = 2m + 2$ are even. It is well known (and follows from the explicit formula for $P_{2m}(0)$) that

$$P_{2m}(0) = (-1)^m\, b_m, \qquad b_m := \frac{(2m)!}{2^{2m}(m!)^2} = \frac{\binom{2m}{m}}{4^m}.$$ (38)

Therefore,

$$P_{2m+2}(0) - P_{2m}(0) = (-1)^{m+1} b_{m+1} - (-1)^m b_m = -(-1)^m (b_{m+1} + b_m),$$

so

$$|P_{2m+2}(0) - P_{2m}(0)| = b_{m+1} + b_m \geq b_m.$$ (39)

Moreover, a standard Stirling bound for the central binomial coefficient implies that there exists an absolute constant $c_0 > 0$ such that

$$b_m = \frac{\binom{2m}{m}}{4^m} \geq \frac{c_0}{\sqrt{m}} \qquad \text{for all } m \geq 1.$$ (40)

Combining (37) (with $n = 2m + 1$), (39), and (40) yields, for $m \geq 1$,

$$|a_{2m+1}| = \frac{|P_{2m+2}(0) - P_{2m}(0)|}{2\sqrt{4m+3}} \geq \frac{b_m}{2\sqrt{4m+3}} \geq \frac{c}{m+1},$$

for some absolute constant $c > 0$ (using $\sqrt{4m+3} \leq 3\sqrt{m+1}$ and $b_m \gtrsim 1/\sqrt{m+1}$).

*Tail lower bound.* Since all even coefficients are zero and $|a_{2m+1}| \gtrsim (m+1)^{-1}$, the best $N$-term Legendre approximation (which keeps the $N$ largest coefficients) cannot remove the harmonic tail. Indeed, for any $N \geq 1$,

$$\sigma_N^{\mathcal{L}}(f_\star)_2^2 = \min_{|S|=N} \sum_{n \notin S} |a_n|^2 \geq \sum_{m > N} |a_{2m+1}|^2 \gtrsim \sum_{m > N} \frac{1}{(m+1)^2} \asymp \frac{1}{N}.$$

Taking square roots gives

$$\sigma_N^{\mathcal{L}}(f_\star)_2 \gtrsim N^{-1/2},$$

which proves (27). $\qquad\square$

### B.2. Further comments

**On the structural assumptions (a)–(c).** Assumptions (a)–(b) formalize the usual *multiresolution geometry* of wavelet systems: at scale $j$ one has atoms supported on intervals of length $\asymp 2^{-j}$, with $\asymp 2^j$ shifts, and with a uniformly bounded number of overlaps at each fixed scale. For standard compactly supported orthonormal wavelets $\psi_{j,k}(x) = 2^{j/2}\psi(2^j x - k)$ these properties follow immediately from the support formula $\mathrm{supp}(\psi_{j,k}) \subset [(k+a)2^{-j}, (k+b)2^{-j}]$ (if $\mathrm{supp}(\psi) \subset [a,b]$), implying a uniform overlap bound independent of $j$. For interval-adapted wavelets (periodic, boundary-corrected, or

extension-based constructions), only $O(1)$ additional boundary atoms appear per scale, so the same bounded-overlap property still holds.

Assumption (c) is the standard *cancellation* (vanishing-moment) mechanism: if $f$ is $H^s$-smooth on an interval containing $\text{supp}(\phi_\lambda)$, then $\langle f, \phi_\lambda \rangle$ decays like $2^{-j(s+1/2)}$. For classical wavelets this is obtained by Taylor expansion plus vanishing moments (or equivalent Strang–Fix type conditions) and is a cornerstone in wavelet approximation theory.

**On the Parseval (tight) frame assumption.** The Parseval property is mainly a technical convenience: it lets one control $L^2$ error by coefficient tail energy and makes the nonlinear "keep the $N$ largest coefficients" argument clean. Many practically used redundant wavelet systems are *tight* (or nearly tight) by design, and there are systematic construction principles producing tight wavelet frames with good localization and approximation order. More generally, for a frame with bounds $A, B$ one can rewrite the proof with the same structure, with constants depending on $A, B$ (or by passing to the canonical tight frame via the frame operator), at the expense of heavier notation.

**Beyond compact support: rapidly decaying / continuous wavelet frames.** Compact support is *sufficient* (not necessary) for (a)–(b) and for the $L^1$ scaling estimates used in the "singular tail" argument. For non-compactly supported wavelets (e.g. Meyer-type bandlimited wavelets, Morlet-type wavelets, or other smooth frames), analogous localization statements hold in terms of rapid decay rather than strict support, and the same approximation phenomena persist. However, the proofs typically require replacing support-counting by decay estimates and using more technical machinery (e.g. discretization of continuous transforms / coorbit or atomic decomposition theory) to control overlaps and tails quantitatively.

## C. Approximating step functions on a budget

Therem B.1 outlines a general principle that wavelets approximate better functions which exhibit irregularities (such as jump discontinuity) than polynomials (our toy example was Legendre orthogonal frame). Here we provide experimental verification of this that accompanies the informal treatment in Section 3.3 to compare $N$-term approximations of simple piecewise-constant signals (the simplest class of functions exhibiting an irregularity in the form of jump discontinuity) under a fixed coefficient budget $N$.

### C.1. Test signals

We consider two canonical targets on $[0, 1]$:

$$f_{\text{one}}(t) := \mathbf{1}_{[1/2,\, 1]}(t) \quad \text{(one step)}, \tag{41}$$
$$f_{\text{two}}(t) := a_0\, \mathbf{1}_{[0,t_1)}(t) + a_1\, \mathbf{1}_{[t_1,t_2)}(t) + a_2\, \mathbf{1}_{[t_2,1]}(t), \quad 0 < t_1 < t_2 < 1 \quad \text{(two steps)}. \tag{42}$$

In the experiments that follow we used $t_1 = \frac{1}{3}$, $t_2 = \frac{2}{3}$ and amplitudes $(a_0, a_1, a_2) = (0, 1, 0.3)$.

### C.2. Discretization and error metric

All functions are sampled on a uniform grid

$$t_i = \frac{i}{L-1}, \qquad i = 0, 1, \ldots, L-1,$$

with $L$ sufficiently large (e.g. $L = 2048$). We approximate the continuous $L^2(0, 1)$ error by the discrete quadrature rule

$$\|f - g\|_{L^2(0,1)} \approx \left( \Delta t \sum_{i=0}^{L-1} |f(t_i) - g(t_i)|^2 \right)^{1/2}, \qquad \Delta t = \frac{1}{L-1}. \tag{43}$$

For each method and budget $N$, we report the error $e_N := \|f - \widehat{f}_N\|_2$ computed by (43).

### C.3. Global polynomial baseline: best $N$-term Legendre approximation

We use Let denote the orthonormal Legendre basis of $L^2(0, 1)$, i.e. $\ell_n(t) = \sqrt{2n+1}\, P_n(2t - 1)$. Given coefficients $a_n = \langle f, \ell_n \rangle$, the best $N$-term approximation in an orthonormal basis is obtained by keeping the $N$ largest $|a_n|$ and

discarding the rest:

$$\widehat{f}_N^{\mathcal{L}}(t) = \sum_{n \in S_N} a_n \, \ell_n(t), \qquad S_N = \arg \max_{|S|=N} \sum_{n \in S} |a_n|^2.$$

In practice, we compute $\{a_n\}_{n=0}^{K-1}$ using high-order Gauss–Legendre quadrature on $[0,1]$ (with $K$ and quadrature order chosen large enough for stability), then select the $N$ largest coefficients among $\{0, \ldots, K-1\}$ and synthesize $\widehat{f}_N^{\mathcal{L}}$ on the sampling grid.

## C.4. Wavelet-based methods

We evaluated two wavelet-style approximation families.

**(i) Orthonormal DWT thresholding (e.g. `db6`).**   As a stable baseline, we use a standard orthonormal discrete wavelet transform (DWT) with a compactly supported Daubechies wavelet (default `db6`). We use `python` library `pywavelets`. Let $w$ denote the full collection of wavelet coefficients produced by the DWT. We form an $N$-term approximation by hard-thresholding: keep the $N$ coefficients of largest magnitude and set the rest to zero, then invert the DWT to obtain $\widehat{f}_N^{\mathrm{DWT}}$. Since the DWT is orthonormal (up to boundary handling), this procedure is a standard proxy for best $N$-term wavelet approximation.

**(ii) Continuous-wavelet-type dictionaries from a mother wavelet (redundant frames).**   To probe non-DWT wavelet families (e.g. `mexh`, `morl`, `gaus` derivatives), we build a discrete dictionary by sampling a chosen mother wavelet $\psi$ and applying time-consistent dilation/translation:

$$\phi_{a,b}(t) = a^{-1/2} \, \psi\left(\frac{t-b}{a}\right), \qquad a \in \mathcal{A}, \ b \in \mathcal{B}, \tag{44}$$

followed by discrete $L^2$ normalization on the grid (43), as was outlined in Appendix A.2. The scale set $\mathcal{A}$ is chosen geometrically between a finest scale $a_{\min}$ (a few grid steps) and a coarsest scale $a_{\max}$ (a fraction of the interval length), and $\mathcal{B}$ is a uniform set of shifts in $[0,1]$.

*Coefficient selection.* Because (44) yields a redundant (coherent) dictionary, we do not use naive coefficient thresholding. Instead, we compute an $N$-term approximation via a greedy pursuit (Orthogonal Matching Pursuit with a small ridge refit), producing $\widehat{f}_N^{\mathrm{CWT}}$ using at most $N$ atoms.

## C.5. Hyperparameters and reproducibility

All experiments are determined by the tuple

$$(L, \ N, \ K, \ \text{wavelet family}, \ n_{\mathrm{scales}}, \ n_{\mathrm{shifts}}, \ a_{\min}, \ a_{\max}, \ \text{pursuit settings}).$$

Typical choices in the reported runs were:

- Grid size: $L \in \{1024, 2048, 4096\}$.

- Budgets: $N \in \{32, 64, 128, 256\}$.

- Legendre truncation ceiling: $K \approx 2048$ with sufficiently high quadrature order.

- DWT wavelet: `db6` (with periodization or standard boundary handling).

- CWT dictionary: $n_{\mathrm{scales}} \in [24, 64]$, $n_{\mathrm{shifts}} \in [256, 1024]$, $a_{\min} \approx 2\Delta t$–$4\Delta t$, $a_{\max} \approx 0.1$–$0.25$.

- Pursuit: OMP with ridge parameter in the range $10^{-8}$–$10^{-6}$.

## C.6. Plots

The following plots present the result comparisons of the various approximations of the target functions via Legendre frame and corresponding wavelet frames.

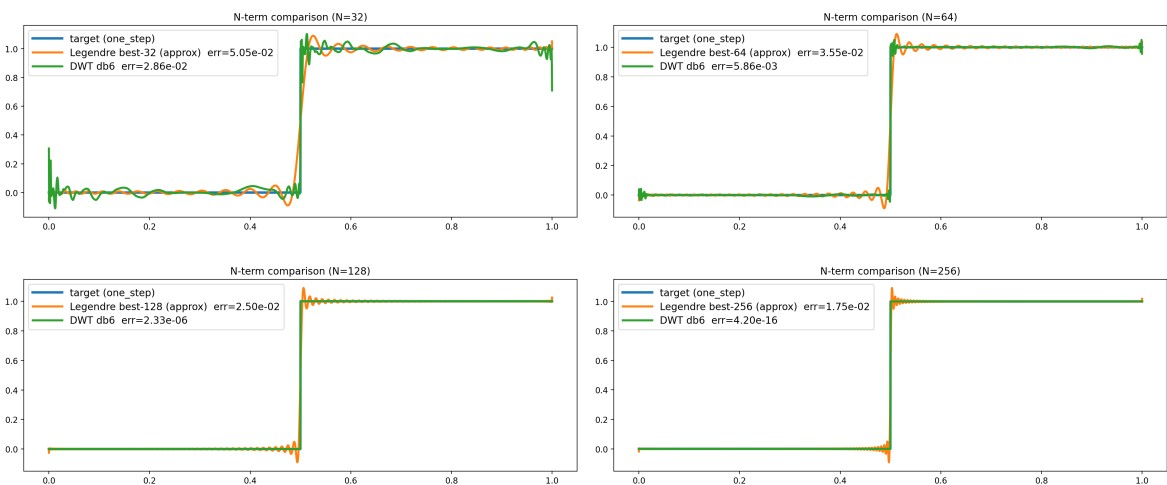

*Figure 7.* Comparison of approximations of $f_{\text{one}}$ target with `db6` and `Legendre` frames on various budgets $N$.

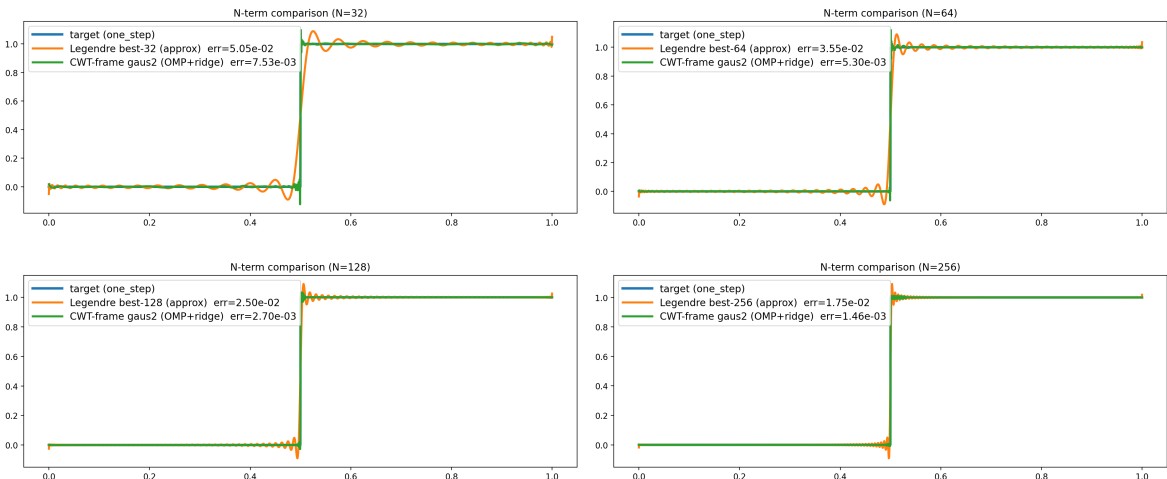

*Figure 8.* Comparison of approximations of $f_{\text{one}}$ target with `gaus2` and `Legendre` frames on various budgets $N$.

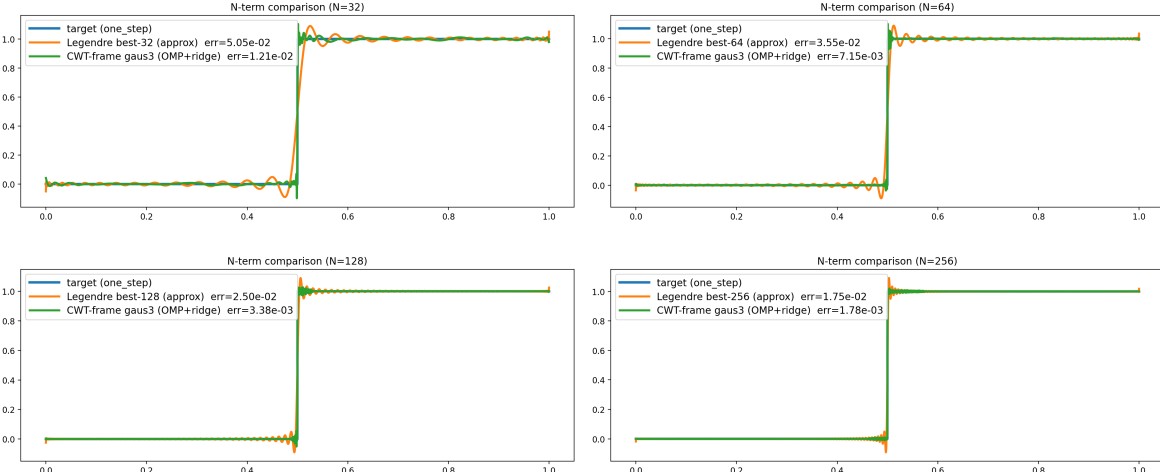

*Figure 9.* Comparison of approximations of $f_{\text{one}}$ target with `gaus3` and `Legendre` frames on various budgets $N$.

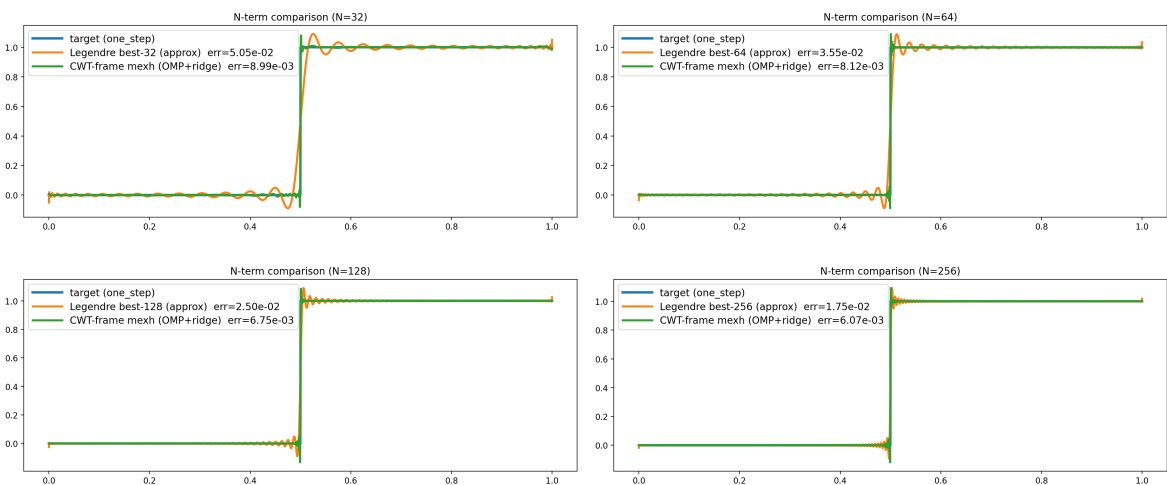

*Figure 10.* Comparison of approximations of $f_{\text{one}}$ target with `mexh` and `Legendre` frames on various budgets $N$.

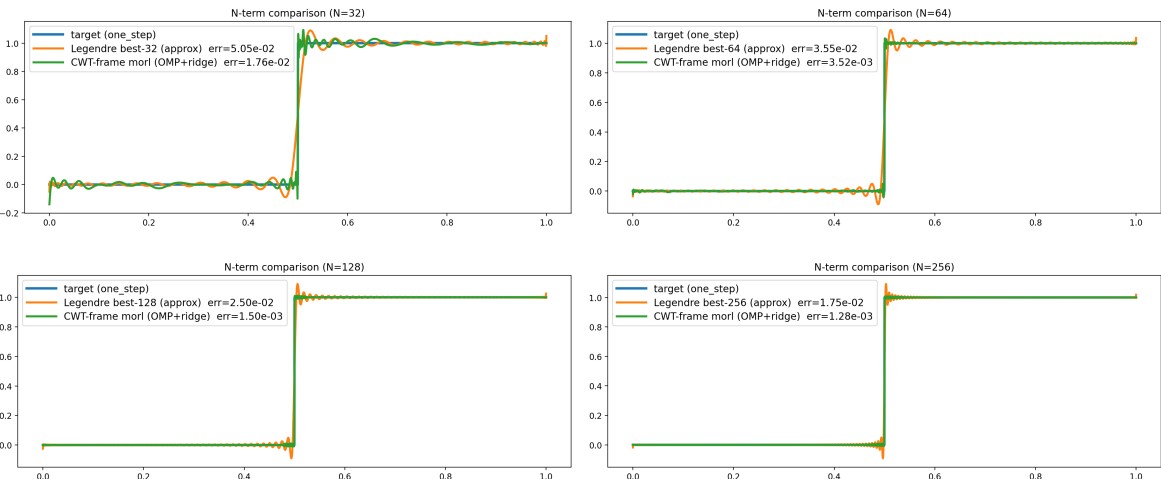

*Figure 11.* Comparison of approximations of $f_{\text{one}}$ target with `morl` and `Legendre` frames on various budgets $N$.

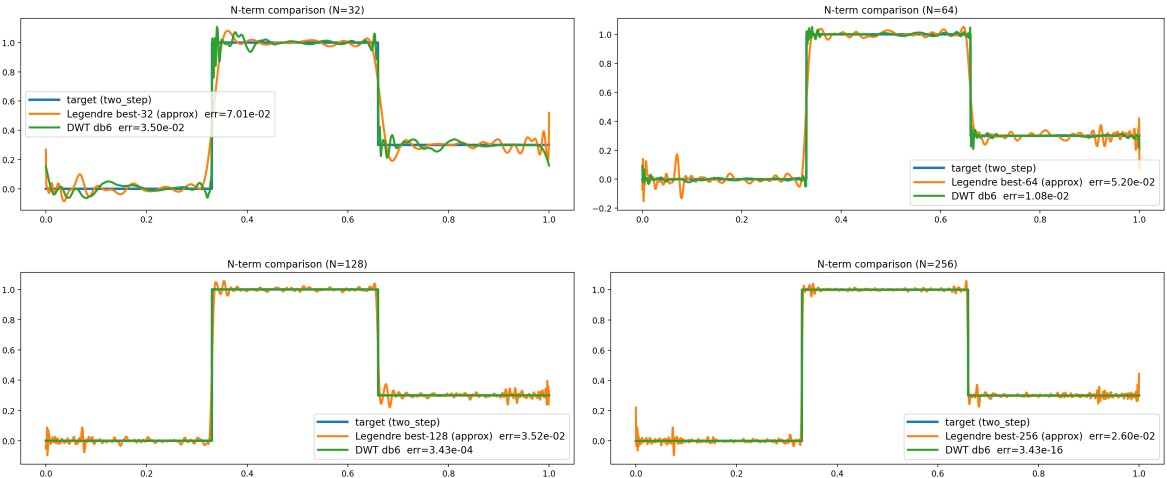

*Figure 12.* Comparison of approximations of $f_{\text{two}}$ target with `db6` and `Legendre` frames on various budgets $N$.

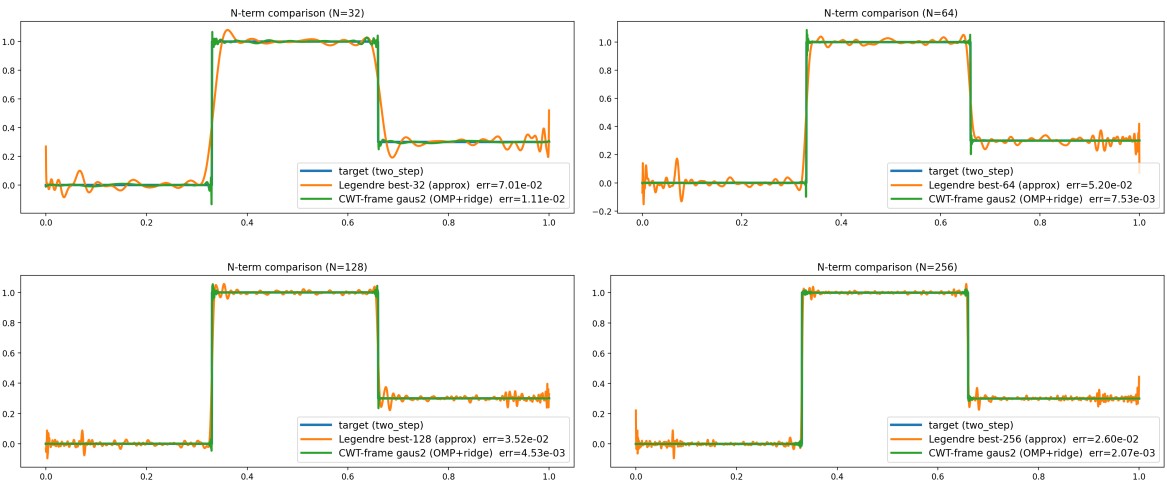

*Figure 13.* Comparison of approximations of $f_{\text{two}}$ target with `gaus2` and `Legendre` frames on various budgets $N$.

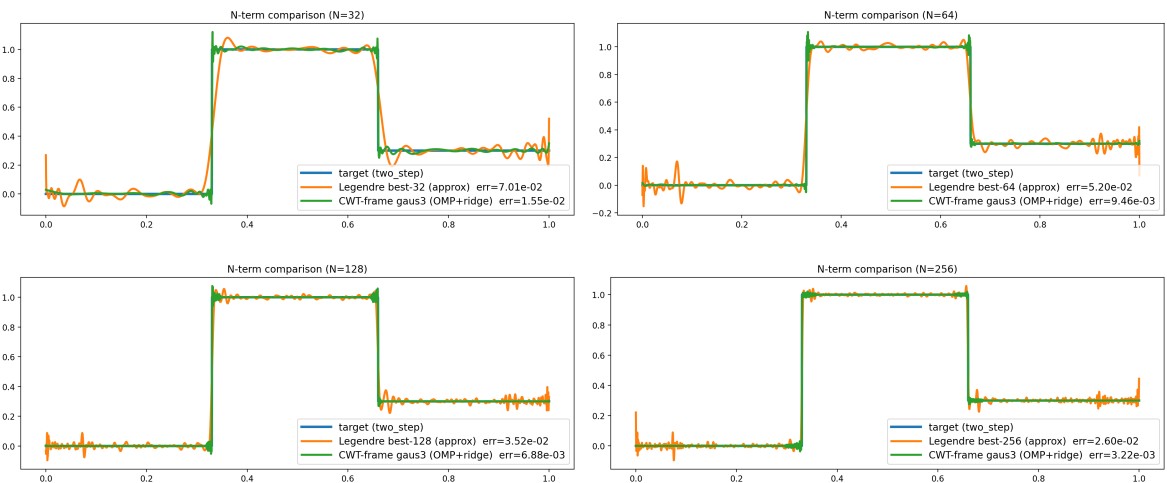

*Figure 14.* Comparison of approximations of $f_{\text{two}}$ target with `gaus3` and `Legendre` frames on various budgets $N$.

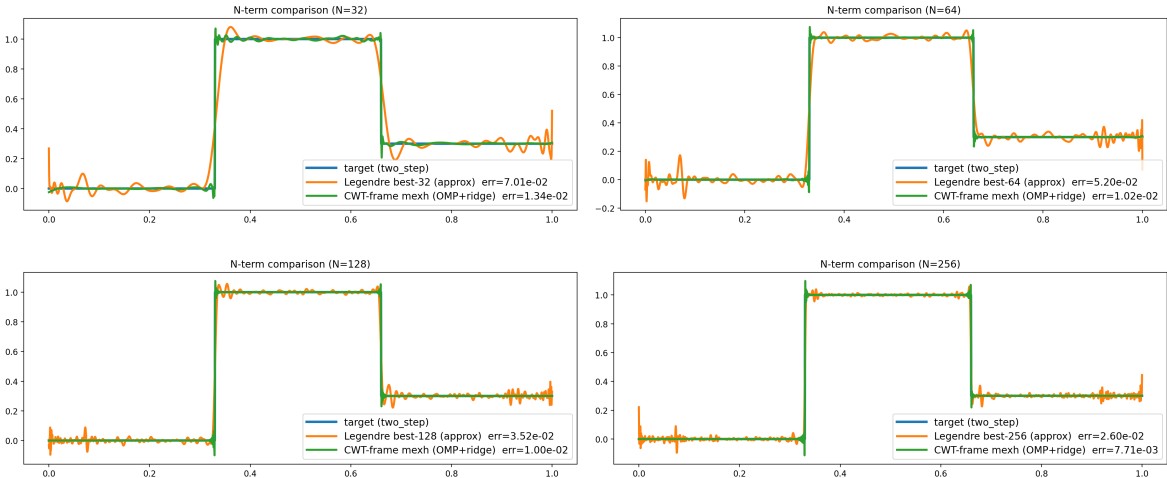

*Figure 15.* Comparison of approximations of $f_{\text{two}}$ target with `mexh` and `Legendre` frames on various budgets $N$.

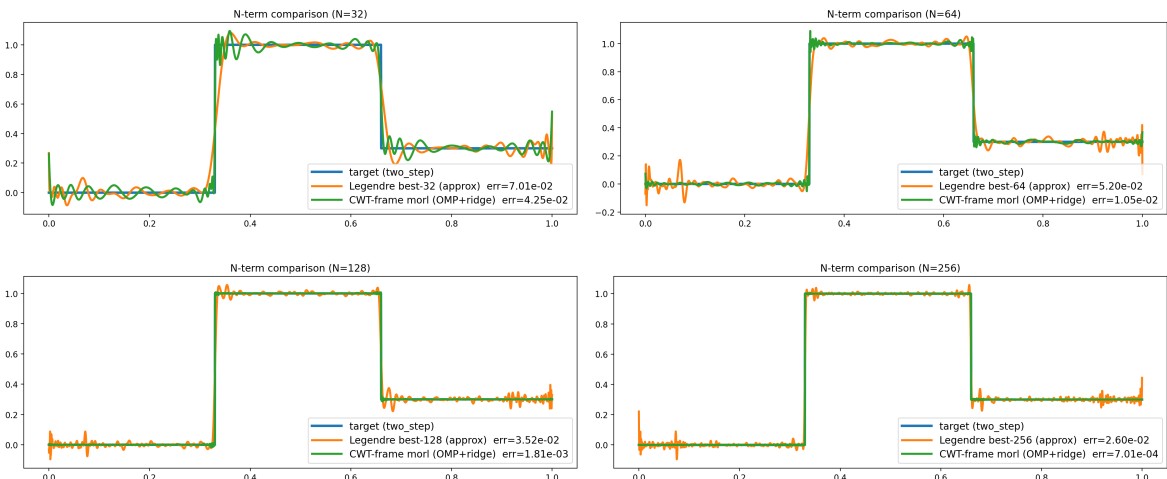

*Figure 16.* Comparison of approximations of $f_{\text{two}}$ target with `morl` and `Legendre` frames on various budgets $N$.

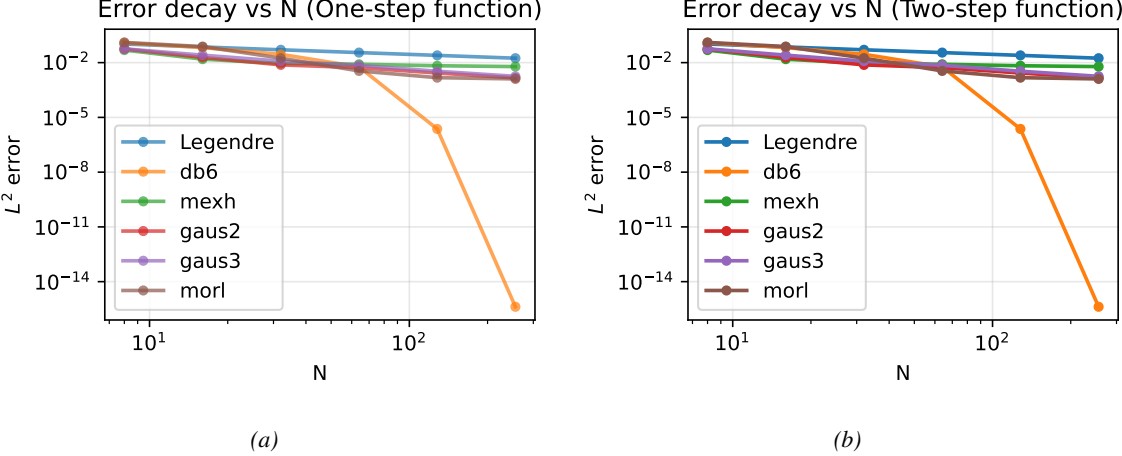

*Figure 17.* Approximation errors on various frames for piecewise constant functions. The $L^2$ approximation error is shown as a function of the number of retained coefficients $N$ on a log–log scale for a one-step function (left) and a two-step function (right). Results are reported for Legendre polynomials and several wavelet-based frames, illustrating differences in convergence behavior and the effect of increasing discontinuity complexity.

# D. Other technical results relevant to the paper

## D.1. Jacobian of the Final State in LTI State-Space Models

We consider the linear discrete-time state-space system

$$h_{t+1} = \bar{A} h_t + \bar{B} x_t, \qquad t = 0, \ldots, T - 1, \tag{45}$$

where $h_t \in \mathbb{R}^N$ denotes the hidden state, $x_t \in \mathbb{R}^D$ the input, and $\bar{A} \in \mathbb{R}^{N \times N}$, $\bar{B} \in \mathbb{R}^{N \times D}$ are fixed matrices. Unrolling the recurrence yields the closed-form expression

$$h_T = \bar{A}^T h_0 + \sum_{t=0}^{T-1} \bar{A}^{T-1-t} \bar{B} x_t. \tag{46}$$

Then the Jacobian of the final hidden state with respect to the input at time $t$ is given by

$$\frac{\partial h_T}{\partial x_t} = \bar{A}^{T-1-t} \bar{B}, \qquad t = 0, \ldots, T - 1. \tag{47}$$

Collecting these Jacobian blocks yields a third-order tensor

$$\mathcal{J} = (J_0, \ldots, J_{T-1}) \in \mathbb{R}^{T \times N \times D}, \tag{48}$$

where $\mathcal{J}_{t,:,:} = \partial h_T / \partial x_t$ denotes the sensitivity of the final hidden state to the input at time $t$.

In the special case of scalar input ($D = 1$), $\bar{B} \in \mathbb{R}^{N \times 1}$ and each $J_t$ reduces to a vector in $\mathbb{R}^N$. Stacking these vectors along the temporal dimension yields the matrix

$$G = \begin{bmatrix} \bar{A}^{T-1} \bar{B}, & \bar{A}^{T-2} \bar{B}, & \ldots, & \bar{B} \end{bmatrix} \in \mathbb{R}^{N \times T}, \tag{49}$$

whose $t$-th column corresponds to the gradient $\partial h_T / \partial x_t$. In other words, each column of G is the impulse response of the SSM at a different time lag.

## D.2. Proof of Lemma 3.1

*Proof.* Let $F \in \mathbb{R}^{N \times L}$ have full row rank and define $S := FF^* \in \mathbb{R}^{N \times N}$. Let $\dot{F} \in \mathbb{R}^{N \times L}$ be fixed, and define

$$A \in \arg \min_{X \in \mathbb{R}^{N \times N}} \|\dot{F} - XF\|_F^2.$$

We prove (i) uniqueness and the closed form $A = \dot{F} F^* S^{-1}$, and (ii) the operator-norm bound.

**Step 1: $S$ is invertible.** Since $F$ has full row rank, $\text{rank}(F) = N$. Hence $S = FF^*$ is symmetric positive definite. Indeed, for any nonzero $x \in \mathbb{R}^N$,

$$x^* S x = x^* F F^* x = \|F^* x\|_2^2 > 0,$$

because $F^* x = 0$ would imply $x$ lies in the left nullspace of $F$, which is trivial when $F$ has full row rank. Therefore $S$ is invertible.

**Step 2: Normal equations and closed form.** Define the objective

$$J(X) := \|\dot{F} - XF\|_F^2 = \text{tr}\left( (\dot{F} - XF)(\dot{F} - XF)^* \right).$$

Expanding the product and using linearity and cyclicity of the trace,

$$J(X) = \text{tr}(\dot{F} \dot{F}^*) - 2\, \text{tr}(XF \dot{F}^*) + \text{tr}(XFF^* X^*).$$

Since $FF^*$ is symmetric, we may differentiate $J$ with respect to $X$ using standard matrix calculus identities:

$$\nabla_X \text{tr}(XM) = M^*, \qquad \nabla_X \text{tr}(XMX^*) = 2XM \quad \text{when } M = M^*.$$

This yields

$$\nabla_X J(X) = -2\,\dot{F}F^* + 2\,X(FF^*).$$

Setting $\nabla_X J(X) = 0$ gives the normal equation

$$X(FF^*) = \dot{F}F^*.$$

By Step 1, $FF^* = S$ is invertible, so the minimizer is unique and equals

$$A = \dot{F}F^*(FF^*)^{-1} = \dot{F}F^*S^{-1}.$$

**Step 3: Operator-norm bound.** By submultiplicativity of the spectral norm,

$$\|A\|_2 = \|\dot{F}F^*S^{-1}\|_2 \leq \|\dot{F}F^*\|_2\,\|S^{-1}\|_2.$$

Since $S$ is symmetric positive definite, $\|S^{-1}\|_2 = 1/\lambda_{\min}(S)$, hence

$$\|A\|_2 \leq \frac{\|\dot{F}F^*\|_2}{\lambda_{\min}(S)}.$$

$\square$

# E. Design Choices of the Wavelet Frames

We build the frame matrices $F \in \mathbb{R}^{N \times L}$ ($N$ atoms of $L$ = sequence length) for Morlet, Gaussian-derivative, Mexican-hat, Daubechies and DPSS following the same high-level recipe:

**(1) Frequency grid $\rightarrow$ scales.** We start choosing pseudo-frequencies $f_k \in [f_{\min}, f_{\max}]$ (log-spaced) as hyperparameters. For each wavelet family, we use its *central frequency* $f_c$ to convert such pseudo-frequencies of choice into a dilation scale, $s_k = \frac{f_c}{f_k}$, so larger $f_k$ produces a smaller $s_k$ and hence a narrower atom.

**(2) Scale of the prototype wavelet.** To generate an atom at scale $s_k$ and center $c$, we sample a *shifted and dilated* copy of the mother wavelet $\psi$ on the length-$L$ grid. Specifically, using the wavelet coordinate grid $x$, we recenter at $x(c)$ and rescale by $s_k$, and then interpolate the resulting waveform onto the discrete scales:

$$\varphi_{k,c}[t] \propto \psi\left(\frac{x(t) - x(c)}{s_k}\right). \tag{50}$$

**(3) Bandwidth-aware shift density.** Different scales have different temporal widths, so we place narrow atoms more densely than wide atoms. In order to do it, we first estimate the prototype time spread $\sigma_0$ (energy-weighted standard deviation of the prototype). This implies a scale-dependent width $\sigma_k \propto s_k\,\sigma_0$. We then choose a hop size proportional to this width:

$$\text{hop}_k \approx \alpha\,\sigma_k \qquad \text{with} \qquad \alpha \approx 0.75. \tag{51}$$

In order to have exactly $N$ atoms overall, we interpret $L/\text{hop}_k$ as a *desired* number of centers per scale and convert these values into integer allocations $\{n_k\}$ that satisfy $\sum_k n_k = N$. We first normalize the desired counts into proportions, set $n_k$ to the integer part of the resulting value, and then distribute the remaining atoms to the scales with the largest remainders. Finally, we place the $n_k$ centers uniformly over $[0, L-1]$ for each scale.

**(4) Normalization.** Each atom is normalized to unit energy, $\|\varphi\|_2 = 1$.

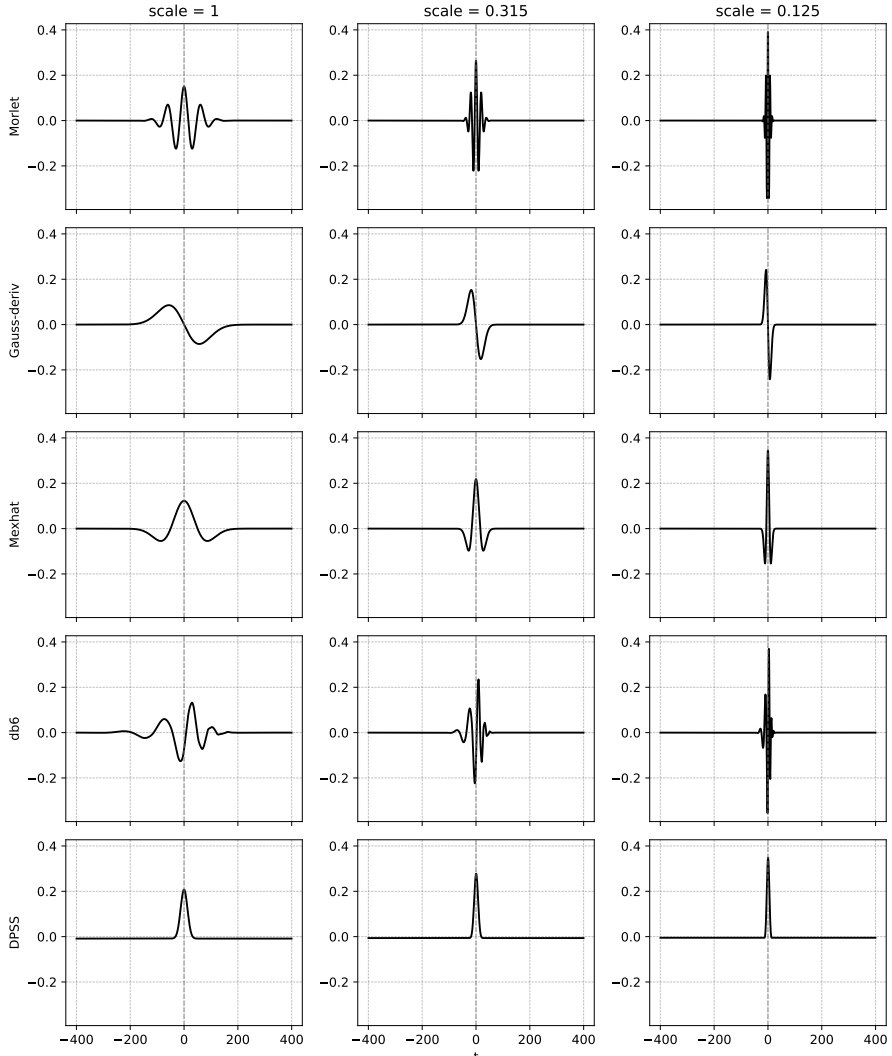

*Figure 18.* Representative atoms for Morlet, Gaussian-derivative, Mexican hat, Daubechies, and DPSS (rows) at fixed scales $s \in \{1, 0.315, 0.125\}$ (columns), which control temporal width. All atoms are $\ell_2$-normalized; therefore, as a consequence of energy normalization, peak magnitudes vary across scales.

## F. Experimental Details

### F.1. ECG Classification

**Dataset details.** We conduct experiments on the publicly available PTB-XL benchmark dataset (Wagner et al., 2020), which consists of 21,837 ten-second recordings from standard 12-lead ECGs acquired from 18,885 unique patients. Each recording is independently reviewed by up to two cardiologists and assigned one or more labels from a vocabulary of 71 ECG statements defined under the SCP-ECG standard. The annotations are grouped into three high-level types: diagnostic, form, and rhythm. Diagnostic labels additionally follow a hierarchical structure with 5 coarse-grained superclasses and 24 fine-grained subclasses. Leveraging this structure, there are six prediction tasks at different semantic resolutions: **all** (all available statements), **diag** (all diagnostic labels across hierarchy levels), **sub-diag** (diagnostic subclasses), **super-diag** (diagnostic superclasses), **form**, and **rhythm**. The super-diag task is posed as a multi-class classification problem, whereas the remaining tasks are formulated as multi-label classification.

**Training details.** We train WaveSSM for 100 epochs using the AdamW optimizer. The model comprises 6 layers, each containing 256 SSMs per layer of order $N = 64$. We follow the training pipeline from https://github.com/state-

spaces/s4/tree/main.

## F.2. Informer Time-series Forecasting

**Dataset details.** Our experiments follow the standard Informer time-series forecasting benchmark (Zhou et al., 2021), which include electricity transformer temperature datasets (ETTh1, ETTh2, ETTm1) with both hourly and minutely measurements. We evaluate performance under long-horizon forecasting regimes with prediction lengths of 24, 48, 168, 336, 720, and 960 time steps. Experiments are conducted in both univariate and multivariate settings: univariate forecasting predicts a single target series, while multivariate forecasting leverages all available variables as inputs. In addition to the ETT datasets, we also consider other real-world multivariate time series from the Informer benchmark, including weather and electricity load (ECL) data.

**Training details.** We train WaveSSM on both univariate and multivariatete variants using the AdamW optimizer and a shallow architecture with two layers. Consistent with prior work, the sequence length, prediction length, and label length hyperparameters are taken from https://github.com/zhouhaoyi/Informer2020/tree/main/scripts. As the model is quite simple to overfit, we perform early stopping.

## F.3. Speech Commands

**Dataset details.** We use the Speech Commands dataset (Warden, 2018), a benchmark corpus for limited-vocabulary keyword spotting. The dataset comprises 105,829 short audio recordings from 35 spoken-word classes, collected from 2,618 distinct speakers. Each recording is provided as a single-channel file up to 1 second in duration, encoded at a 16 kHz sampling rate. Additionally, as introduced in (Gu et al., 2022a), we test zero-shot performance when the signal is undersampled to 8kHz.

**Training details.** We train WaveSSM using the same overall training protocol as S4, with 128 SSMs per layer across 6 layers. The reported results are obtained by retraining both models under an identical configuration, changing only the initialization of the $A$ and $B$ matrices.

## F.4. Long Range Arena

**Dataset details.** The Long Range Arena (LRA) benchmark is a collection of sequence classification tasks designed to evaluate a model's ability to capture long-range dependencies across diverse input modalities. It comprises the following six tasks:

- **ListOps.** This dataset consists of synthetically generated mathematical expressions formed by recursively applying operators such as $\min$ and $\max$ to single-digit integers (0–9). Expressions are written in prefix notation with explicit parentheses. Inputs are tokenized at the character level and represented using one-hot vectors over a vocabulary of 17 symbols, where all operators and opening brackets are mapped to a shared token. Since expression lengths vary, sequences are padded with a special symbol to a maximum length of 2048. The objective is to predict the final integer result, which yields a classification problem of 10 classes.

- **Text.** This task corresponds to sentiment classification on the IMDb dataset. Each movie review is represented as a sequence of character tokens, and the goal is to determine whether the sentiment is positive or negative. Characters are one-hot encoded using a vocabulary of 129 symbols, and sequences are padded to a fixed length of 4096.

- **Retrieval.** Based on the ACL Anthology Network corpus, this task presents pairs of citation strings encoded as character sequences. The model must decide whether the two citations refer to the same underlying publication. Characters are represented with one-hot vectors over 97 tokens. Variable-length sequences are padded to a maximum of 4000 tokens, and a dedicated end-of-sequence marker is appended. The output is binary: matching or non-matching.

- **Image.** This task reformulates the CIFAR-10 image classification problem as a one-dimensional sequence modeling task. RGB images are first converted to grayscale and normalized to have zero mean and unit variance across the dataset. The resulting $32 \times 32$ images are flattened into sequences of length 1024, and the model predicts one of the 10 CIFAR-10 classes.

- **Pathfinder.** Each example is a $32 \times 32$ grayscale image containing two marked points and a collection of dashed line segments. Pixel values are normalized to the range $[-1, 1]$. The task is to determine whether a continuous dashed path exists between the two marked points.

- **PathX.** This task is a more challenging extension of Pathfinder, operating on higher-resolution images. In the `PathX-128` setting, inputs are $128 \times 128$ grayscale images, resulting in substantially longer input sequences and increased difficulty.

**Training details.** For all Long Range Arena tasks, we train WaveSSM with 6 layers, using task-dependent hidden and state dimensions. Depending on the task, we use batch normalization or layer normalization, enable pre-normalization when required, and apply dropout only where specified; the learning rate, batch size, weight decay, and number of training epochs are tuned per task. We follow the training pipeline from https://github.com/state-spaces/s4/tree/main.

All architectural choices and optimization hyperparameters are summarized in Table 5.

*Table 5.* Hyperparameters used for the reported results, shared across WaveSSM and S4. L is the number of layers; H, the embedding size; N, the order of the SSM; Dropout, the dropout rate; LR, the learning rate; BS, the batch size; Epochs, the training epochs; WD, weight decay; and $(\Delta_{\min}, \Delta_{\max})$, the range for the discretization step-size.

| Task | $L$ | $H$ | $N$ | Norm | Pre-norm | Dropout | LR | BS | Epoch | WD | $(\Delta_{\min}, \Delta_{\max})$ |
|---|---|---|---|---|---|---|---|---|---|---|---|
| ListOps | 6 | 256 | 64 | BN | False | 0 | 0.01 | 50 | 40 | 0.05 | (0.001, 0.1) |
| Text | 6 | 256 | 64 | BN | True | 0 | 0.01 | 16 | 32 | 0.05 | (0.001, 0.1) |
| Retrieval | 6 | 256 | 64 | BN | True | 0 | 0.01 | 64 | 20 | 0.05 | (0.001, 0.1) |
| Image | 6 | 512 | 64 | LN | False | 0.1 | 0.01 | 50 | 200 | 0.05 | (0.001, 0.1) |
| Pathfinder | 6 | 256 | 128 | BN | True | 0 | 0.001 | 64 | 200 | 0.03 | (0.001, 0.1) |
| PathX | 6 | 256 | 128 | BN | True | 0 | 0.0005 | 32 | 50 | 0.05 | (0.001, 0.1) |
| SC35 | 6 | 128 | 64 | BN | True | 0 | 0.001 | 16 | 40 | 0.05 | (0.001, 0.1) |
| ECL | 2 | 128 | 64 | LN | True | 0.25 | 0.01 | 50 | 10 | 0 | (0.001, 0.1) |
| ETTH | 2 | 128 | 64 | LN | True | 0.25 | 0.01 | 50 | 10 | 0 | (0.001, 0.1) |
| ETTM | 2 | 128 | 64 | LN | True | 0.25 | 0.01 | 50 | 10 | 0 | (0.001, 0.1) |
| Weather | 2 | 128 | 64 | LN | True | 0.25 | 0.01 | 50 | 10 | 0 | (0.001, 0.1) |
| PTB-XL | 6 | 256 | 64 | BN | True | 0.1 | 0.01 | 16 | 100 | 0.05 | (0.001, 0.1) |

# G. Extended Experimental Results

We have conducted complementary ablations, always having in mind Remark 6.1 and preserving the "same budget" in all cases. Particularly, we ablate the $n_{scale}$ and the linear or log-spaced pseudo-frequency grid $[f_{min}, f_{max}]$ from where the scaling factors are sampled. The experiments conducted on MorletS and MorletT wavelets on Cifar and Listops datasets are available on Table 9 and Table 10, respectively. Results show the method is robust to the selection of those hyperparameters. A general rule of thumb that is robust enough and that we followed in the paper is to set, and set a wide pseudo-frequency default interval to [0.25-2] and choose from it based on.

*Table 6.* Univariate Informer's long sequence time-series forecasting results (MSE ↓).

| Model | ETTh$_1$ | | | | | ETTh$_2$ | | | | | ETTm$_1$ | | | | |
|---|---|---|---|---|---|---|---|---|---|---|---|---|---|---|---|
| | 24 | 48 | 168 | 336 | 720 | 24 | 48 | 168 | 336 | 720 | 24 | 48 | 96 | 288 | 672 |
| S4 | 0.061[1] | 0.079[1] | 0.104[2] | 0.080[1] | 0.116[3] | 0.095[3] | 0.191 | 0.167 | 0.189 | 0.187 | 0.024 | 0.051 | 0.086 | 0.160[2] | 0.292 |
| Informer | 0.098 | 0.158 | 0.183 | 0.222 | 0.269 | 0.093[1] | 0.155 | 0.232 | 0.263 | 0.277 | 0.030 | 0.069 | 0.194 | 0.401 | 0.512 |
| LogTrans | 0.103 | 0.167 | 0.207 | 0.230 | 0.273 | 0.102 | 0.169 | 0.246 | 0.267 | 0.303 | 0.065 | 0.078 | 0.199 | 0.411 | 0.598 |
| Reformer | 0.222 | 0.284 | 1.522 | 1.860 | 2.112 | 0.263 | 0.458 | 1.029 | 1.668 | 2.030 | 0.095 | 0.249 | 0.920 | 1.108 | 1.793 |
| LSTMa | 0.114 | 0.193 | 0.236 | 0.590 | 0.683 | 0.155 | 0.190 | 0.385 | 0.558 | 0.640 | 0.121 | 0.305 | 0.287 | 0.524 | 1.064 |
| DeepAR | 0.107 | 0.162 | 0.239 | 0.445 | 0.658 | 0.098 | 0.163 | 0.255 | 0.604 | 0.429 | 0.091 | 0.219 | 0.364 | 0.948 | 2.437 |
| ARIMA | 0.108 | 0.175 | 0.396 | 0.468 | 0.659 | 3.554 | 3.190 | 2.800 | 2.753 | 2.878 | 0.090 | 0.179 | 0.272 | 0.462 | 0.639 |
| Prophet | 0.115 | 0.168 | 1.224 | 1.549 | 2.735 | 0.199 | 0.304 | 2.145 | 2.096 | 3.355 | 0.120 | 0.133 | 0.194 | 0.452 | 2.747 |
| WaveSSM$_{MorletS}$ | 0.077 | 0.148 | 0.134 | 0.153 | 0.114[2] | 0.093[1] | 0.139[3] | 0.165[3] | 0.172 | 0.171 | 0.017 | 0.027[3] | 0.093 | 0.190 | 0.261 |
| WaveSSM$_{GaussS}$ | 0.089 | 0.146 | 0.128 | 0.135 | 0.144 | 0.105 | 0.143 | 0.169 | 0.186 | 0.169 | 0.015[2] | 0.027[3] | 0.102 | 0.183 | 0.245[2] |
| WaveSSM$_{MexhatS}$ | 0.077 | 0.134 | 0.134 | 0.149 | 0.128 | 0.094[2] | 0.140 | 0.173 | 0.171[3] | 0.180 | 0.015[2] | 0.028 | 0.076[2] | 0.175 | 0.247 |
| WaveSSM$_{dpssS}$ | 0.083 | 0.121[3] | 0.102[1] | 0.128 | 0.154 | 0.097 | 0.139[3] | 0.168 | 0.190 | 0.167 | 0.018 | 0.029 | 0.095 | 0.168[3] | 0.260 |
| WaveSSM$_{dpS}$ | 0.076 | 0.139 | 0.141 | 0.141 | 0.130 | 0.095 | 0.141 | 0.167 | 0.200 | 0.183 | 0.016[3] | 0.027 | 0.102 | 0.205 | 0.264 |
| WaveSSM$_{MorletT}$ | 0.083 | 0.138 | 0.141 | 0.131 | 0.135 | 0.097 | 0.141 | 0.173 | 0.191 | 0.156[2] | 0.015[2] | 0.024[1] | 0.092 | 0.168[3] | 0.293 |
| WaveSSM$_{GaussT}$ | 0.084 | 0.111[2] | 0.112[3] | 0.112[2] | 0.102[1] | 0.124 | 0.151 | 0.174 | 0.212 | 0.196 | 0.015[2] | 0.032 | 0.107 | 0.189 | 0.256 |
| WaveSSM$_{MexhatT}$ | 0.065[3] | 0.111[2] | 0.145 | 0.126[3] | 0.136 | 0.093[1] | 0.136[2] | 0.162[2] | 0.162[1] | 0.163[3] | 0.015[2] | 0.030 | 0.074[1] | 0.198 | 0.245[2] |
| WaveSSM$_{dpssT}$ | 0.063[2] | 0.146 | 0.124 | 0.145 | 0.138 | 0.100 | 0.136[2] | 0.160[1] | 0.167[2] | 0.155[1] | 0.014[1] | 0.028 | 0.081[3] | 0.159[1] | 0.231[1] |
| WaveSSM$_{dpT}$ | 0.076 | 0.147 | 0.141 | 0.153 | 0.119 | 0.093[1] | 0.134[1] | 0.172 | 0.189 | 0.179 | 0.014[1] | 0.025[2] | 0.086 | 0.191 | 0.246[3] |

| Model | Weather | | | | | ECL | | | | |
|---|---|---|---|---|---|---|---|---|---|---|
| | 24 | 48 | 168 | 336 | 720 | 48 | 168 | 336 | 720 | 960 |
| S4 | 0.125 | 0.181 | 0.198 | 0.300 | 0.245 | 0.222 | 0.331 | 0.328 | 0.428 | 0.432 |
| Informer | 0.117 | 0.178 | 0.266 | 0.297 | 0.359 | 0.239 | 0.447 | 0.489 | 0.540 | 0.582 |
| LogTrans | 0.136 | 0.206 | 0.309 | 0.359 | 0.388 | 0.280 | 0.454 | 0.514 | 0.558 | 0.624 |
| Reformer | 0.231 | 0.328 | 0.654 | 1.792 | 2.087 | 0.971 | 1.671 | 3.528 | 4.891 | 7.019 |
| LSTMa | 0.131 | 0.190 | 0.341 | 0.456 | 0.866 | 0.493 | 0.723 | 1.212 | 1.511 | 1.545 |
| DeepAR | 0.128 | 0.203 | 0.293 | 0.585 | 0.499 | 0.204 | 0.315 | 0.414 | 0.563 | 0.657 |
| ARIMA | 0.219 | 0.273 | 0.503 | 0.728 | 1.062 | 0.879 | 1.032 | 1.136 | 1.251 | 1.370 |
| Prophet | 0.302 | 0.445 | 2.441 | 1.987 | 3.859 | 0.524 | 2.725 | 2.246 | 4.243 | 6.901 |
| WaveSSM$_{MorletS}$ | 0.097[3] | 0.146 | 0.188[3] | 0.209 | 0.241 | 0.191 | 0.244 | 0.267[1] | 0.274 | 0.283 |
| WaveSSM$_{GaussS}$ | 0.103 | 0.151 | 0.190 | 0.217 | 0.218[1] | 0.188 | 0.249 | 0.282 | 0.252[1] | 0.273[2] |
| WaveSSM$_{MexhatS}$ | 0.097[3] | 0.147 | 0.196 | 0.206[2] | 0.233 | 0.189 | 0.254 | 0.278[3] | 0.287 | 0.283 |
| WaveSSM$_{dpssS}$ | 0.102 | 0.149 | 0.199 | 0.215 | 0.232 | 0.189 | 0.238[2] | 0.269[2] | 0.286 | 0.268[1] |
| WaveSSM$_{dpS}$ | 0.096[2] | 0.147 | 0.195 | 0.206[2] | 0.225 | 0.186 | 0.236[1] | 0.292 | 0.330 | 0.301 |
| WaveSSM$_{MorletT}$ | 0.095[1] | 0.139[1] | 0.192 | 0.228 | 0.233 | 0.182[3] | 0.244 | 0.298 | 0.267[2] | 0.316 |
| WaveSSM$_{GaussT}$ | 0.123 | 0.181 | 0.201 | 0.219 | 0.231 | 0.213 | 0.278 | 0.314 | 0.295 | 0.312 |
| WaveSSM$_{MexhatT}$ | 0.095[1] | 0.140[2] | 0.182[1] | 0.213 | 0.220[2] | 0.196 | 0.242[3] | 0.281 | 0.270[3] | 0.310 |
| WaveSSM$_{dpssT}$ | 0.100 | 0.144 | 0.190 | 0.208[3] | 0.223[3] | 0.173[1] | 0.238[2] | 0.282 | 0.297 | 0.277[3] |
| WaveSSM$_{dpT}$ | 0.095[1] | 0.142[3] | 0.185[2] | 0.202[1] | 0.229 | 0.174[2] | 0.238[2] | 0.283 | 0.279 | 0.313 |

*Table 7.* Multivariate Informer's long sequence time-series forecasting results (MSE ↓).

| Model | | ETTh$_1$ | | | | | | ETTh$_2$ | | | | | | ETTm$_1$ | | | | |
|---|---|---|---|---|---|---|---|---|---|---|---|---|---|---|---|---|---|---|
| | | 24 | 48 | 168 | 336 | 720 | | 24 | 48 | 168 | 336 | 720 | | 24 | 48 | 96 | 288 | 672 |
| S4 | | 0.525 | 0.641 | 0.980 | 1.407 | 1.162 | | 0.871 | 1.240 | 2.580 | 1.980[1] | 2.650[3] | | 0.426 | 0.580 | 0.699 | 0.824 | 0.846 |
| Informer | | 0.577 | 0.685 | 0.931 | 1.128 | 1.215 | | 0.720 | 1.457 | 3.489 | 2.723 | 3.467 | | 0.323[1] | 0.494 | 0.678 | 1.056 | 1.192 |
| LogTrans | | 0.686 | 0.766 | 1.002 | 1.362 | 1.397 | | 0.828 | 1.806 | 4.070 | 3.875 | 3.913 | | 0.419 | 0.507 | 0.768 | 1.462 | 1.669 |
| Reformer | | 0.991 | 1.313 | 1.824 | 2.117 | 2.415 | | 1.531 | 1.871 | 4.660 | 4.028 | 5.381 | | 0.724 | 1.098 | 1.433 | 1.820 | 2.187 |
| LSTMa | | 0.650 | 0.702 | 1.212 | 1.424 | 1.960 | | 1.143 | 1.671 | 4.117 | 3.434 | 3.963 | | 0.621 | 1.392 | 1.339 | 1.740 | 2.736 |
| LSTnet | | 1.293 | 1.456 | 1.997 | 2.655 | 2.143 | | 2.742 | 3.567 | 3.242 | 2.544 | 4.625 | | 1.968 | 1.999 | 2.762 | 1.257 | 1.917 |
| WaveSSM$_{MorletS}$ | | 0.432 | 0.431[1] | 0.830[3] | 1.057 | 1.062 | | 0.578 | 1.254 | 3.121 | 2.575 | 2.894 | | 0.355[3] | 0.442[3] | 0.413[2] | 0.505[1] | 0.640[2] |
| WaveSSM$_{GaussS}$ | | 0.490 | 0.528 | 0.892 | 0.956[2] | 1.078 | | 0.574 | 1.469 | 2.811 | 2.554 | 2.856 | | 0.356 | 0.444 | 0.433 | 0.507[2] | 0.616[1] |
| WaveSSM$_{MexhatS}$ | | 0.408[1] | 0.432[2] | 0.763[1] | 1.078 | 1.089 | | 0.567[3] | 1.179[2] | 2.752 | 2.530[3] | 2.873 | | 0.350[2] | 0.434 | 0.425[3] | 0.516 | 0.674 |
| WaveSSM$_{dpssS}$ | | 0.474 | 0.495 | 0.835 | 0.964[3] | 1.068 | | 0.615 | 1.186[3] | 2.549[3] | 2.635 | 2.735 | | 0.367 | 0.461 | 0.428 | 0.536 | 0.641[3] |
| WaveSSM$_{MorletT}$ | | 0.442 | 0.445[3] | 0.838 | 1.056 | 1.038[2] | | 0.435[1] | 1.231 | 2.753 | 2.677 | 2.724 | | 0.360 | 0.416[1] | 0.425[3] | 0.508[3] | 0.667 |
| WaveSSM$_{GaussT}$ | | 0.714 | 0.815 | 1.112 | 1.123 | 1.109 | | 1.741 | 2.075 | 2.251[1] | 2.459[2] | 2.481[1] | | 0.367 | 0.448 | 0.427 | 0.600 | 0.749 |
| WaveSSM$_{MexhatT}$ | | 0.429[3] | 0.445[3] | 0.806[2] | 1.072 | 1.069[3] | | 0.505[2] | 0.989[1] | 2.633 | 2.698 | 2.611[2] | | 0.350[2] | 0.480 | 0.438 | 0.526 | 0.751 |
| WaveSSM$_{dpssT}$ | | 0.412[2] | 0.463 | 0.854 | 0.928[1] | 0.975[1] | | 0.702 | 1.276 | 2.457[2] | 2.679 | 2.703 | | 0.359 | 0.461 | 0.393[1] | 0.529 | 0.722 |

| Model | | Weather | | | | | | ECL | | | | |
|---|---|---|---|---|---|---|---|---|---|---|---|---|
| | | 24 | 48 | 168 | 336 | 720 | | 48 | 168 | 336 | 720 | 960 |
| S4 | | 0.334 | 0.406 | 0.525 | 0.531[3] | 0.578 | | 0.255[2] | 0.283 | 0.292 | 0.289[2] | 0.299[3] |
| Informer | | 0.335 | 0.395 | 0.608 | 0.702 | 0.831 | | 0.344 | 0.368 | 0.381 | 0.406 | 0.460 |
| LogTrans | | 0.435 | 0.426 | 0.727 | 0.754 | 0.885 | | 0.355 | 0.368 | 0.373 | 0.409 | 0.477 |
| Reformer | | 0.655 | 0.729 | 1.318 | 1.930 | 2.726 | | 1.404 | 1.515 | 1.601 | 2.009 | 2.141 |
| LSTMa | | 0.546 | 0.829 | 1.038 | 1.657 | 1.536 | | 0.486 | 0.574 | 0.886 | 1.676 | 1.591 |
| LSTnet | | 0.615 | 0.660 | 0.748 | 0.782 | 0.851 | | 0.369 | 0.394 | 0.419 | 0.556 | 0.605 |
| WaveSSM$_{MorletS}$ | | 0.320 | 0.396 | 0.487 | 0.537 | 0.539[3] | | 0.262 | 0.288 | 0.287[2] | 0.292 | 0.310 |
| WaveSSM$_{GaussS}$ | | 0.335 | 0.405 | 0.490 | 0.537 | 0.581 | | 0.265 | 0.276[1] | 0.295 | 0.290[3] | 0.297 |
| WaveSSM$_{MexhatS}$ | | 0.319 | 0.391[3] | 0.482[2] | 0.540 | 0.538[2] | | 0.259[3] | 0.277[2] | 0.304 | 0.304 | 0.299[3] |
| WaveSSM$_{dpssS}$ | | 0.324 | 0.397 | 0.484[3] | 0.531[3] | 0.546 | | 0.262 | 0.284 | 0.300 | 0.294 | 0.295[2] |
| WaveSSM$_{MorletT}$ | | 0.318[3] | 0.388[2] | 0.485 | 0.524[1] | 0.534[1] | | 0.266 | 0.293 | 0.295 | 0.306 | 0.309 |
| WaveSSM$_{GaussT}$ | | 0.374 | 0.468 | 0.520 | 0.556 | 0.558 | | 0.277 | 0.283 | 0.279[1] | 0.284[1] | 0.286[1] |
| WaveSSM$_{MexhatT}$ | | 0.312[1] | 0.385[1] | 0.480[1] | 0.527[2] | 0.534[1] | | 0.267 | 0.279 | 0.291[3] | 0.351 | 0.327 |
| WaveSSM$_{dpssT}$ | | 0.315[2] | 0.396 | 0.487 | 0.524[1] | 0.541 | | 0.251[1] | 0.278[3] | 0.301 | 0.296 | 0.303 |

*Table 8.* Extended comparison on LRA.

| Model | ListOps (2,048) | Text (4,096) | Retrieval (4,000) | Image (1,024) | Pathfinder (1,024) | PathX-128 (16,384) |
|---|---|---|---|---|---|---|
| Transformer (Vaswani et al., 2017a) | 36.37 | 64.27 | 57.46 | 42.44 | 71.40 | ✗ |
| Reformer (Kitaev et al., 2020) | 37.27 | 56.10 | 53.40 | 38.07 | 68.50 | ✗ |
| BigBird (Zaheer et al., 2020) | 36.05 | 64.02 | 59.29 | 40.83 | 74.87 | ✗ |
| Linear Trans. (Katharopoulos et al., 2020) | 16.13 | 65.90 | 53.09 | 42.34 | 75.30 | ✗ |
| Performer (Choromanski et al., 2021) | 18.01 | 65.40 | 53.82 | 42.77 | 77.05 | ✗ |
| MEGA (Ma et al., 2023) | 63.14[1] | 90.43[1] | 91.25 | 90.44[1] | 96.01[1] | 97.98[2] |
| DSS (Gupta et al., 2022) | 60.60 | 84.80 | 87.80 | 85.70 | 84.60 | 87.80 |
| S4-FouT (Gu et al., 2022b) | 57.88 | 86.34 | 89.66 | 89.07 | 94.46 | ✗ |
| S4-LegS (Gu et al., 2022b) | 59.60 | 86.82 | 90.90 | 88.65 | 94.20 | 96.35 |
| Liquid-S4 (Hasani et al., 2023) | 62.75[2] | 89.02 | 91.20 | 89.50 | 94.80 | 96.66 |
| S5-Inv (Smith et al., 2023) | 60.07 | 87.77 | 91.26 | 86.41 | 93.42 | 97.54[3] |
| S5-Lin (Smith et al., 2023) | 59.98 | 88.15 | 91.31 | 86.05 | 94.31 | 65.60 |
| S5 (Smith et al., 2023) | 62.15[3] | 89.31 | 91.40 | 88.00 | 95.33[2] | 98.58[1] |
| LRU (Orvieto et al., 2023) | 60.2 | 89.4[2] | 89.9 | 89.0 | 95.1[3] | 94.2 |
| WaveSSM$_{MorletS}$ | 60.88 | 89.11 | 91.97[1] | 89.26 | 94.94 | ✗ |
| WaveSSM$_{GaussS}$ | 59.62 | 88.06 | 91.19 | 89.68 | ✗ | ✗ |
| WaveSSM$_{MexhatS}$ | 60.13 | 89.33[3] | 91.88[2] | 89.35 | ✗ | ✗ |
| WaveSSM$_{dpssS}$ | 61.74 | 88.56 | 91.52 | 89.41 | ✗ | ✗ |
| WaveSSM$_{MorletT}$ | 61.79 | 89.23 | 91.88[2] | 89.86[2] | 73.31 | ✗ |
| WaveSSM$_{GaussT}$ | 60.93 | 87.65 | 90.89 | 87.26 | 94.33 | ✗ |
| WaveSSM$_{MexhatT}$ | 61.34 | 89.05 | 91.68[3] | 89.74[3] | 73.42 | ✗ |
| WaveSSM$_{dpssT}$ | 60.98 | 89.01 | 91.48 | 89.04 | 62.51 | ✗ |

*Table 9.* Ablations on LRA/CIFAR for the proposed Morlet-based frame design. **Left:** accuracy upon the change on the number of scales $n_{\text{scales}}$ in the frame. **Right:** effect of the pseudo-frequency interval $f_k \in [f_{\min}, f_{\max}]$ used to sample the *scale k*; comparison between logarithmic and linear grids.

*(a)* Number of scales ablation

| $n_{\text{scales}}$ | MorletS | MorletT |
|---|---|---|
| 2 | 89.48 | 89.29 |
| 5 | 89.60 | 89.93 |
| 10 | 89.62 | 89.38 |
| 15 | 89.25 | 89.79 |
| 20 | 89.09 | 88.97 |
| 30 | 88.77 | 89.16 |

*(b)* Pseudo-frequency interval ablation

| Scale | Setting | Interval | MorletS | MorletT |
|---|---|---|---|---|
| LOG | narrow low band | [0.25–1] | 89.69 | 89.30 |
| | wider band | [0.25–4] | 89.18 | 89.39 |
| | wider high band | [0.25–10] | 89.37 | 89.56 |
| | shifted upward | [0.5–2] | 89.56 | 89.47 |
| | shifted downward | [0.125–2] | 89.53 | 89.47 |
| LINEAR | narrow low band | [0.25–1] | 89.52 | 88.89 |
| | wider band | [0.25–4] | 89.27 | 88.94 |
| | wider high band | [0.25–10] | 88.61 | 88.60 |
| | shifted upward | [0.5–2] | 89.73 | 89.24 |
| | shifted downward | [0.125–2] | 89.71 | 89.54 |

*Table 10.* Ablations on LRA/Listops for the proposed Morlet-based frame design. **Left:** accuracy upon the change on the number of scales $n_{\text{scales}}$ in the frame. **Right:** effect of the pseudo-frequency interval $f_k \in [f_{\min}, f_{\max}]$ used to sample the *scale k*; comparison between logarithmic and linear grids.

*(a)* Number of scales ablation

| $n_{\text{scales}}$ | MorletS | MorletT |
|---|---|---|
| 2 | 61.84 | 61.79 |
| 5 | 62.29 | 61.59 |
| 10 | 62.85 | 60.08 |
| 15 | 60.98 | 60.68 |
| 20 | 60.63 | 59.22 |
| 30 | 60.88 | 61.79 |

*(b)* Pseudo-frequency interval ablation

| Scale | Setting | Interval | MorletS | MorletT |
|---|---|---|---|---|
| LOG | narrow low band | [0.25–1] | 61.43 | 61.19 |
| | wider band | [0.25–4] | 61.48 | 61.22 |
| | wider high band | [0.25–10] | 61.03 | 60.78 |
| | shifted upward | [0.5–2] | 60.93 | 61.24 |
| | shifted downward | [0.125–2] | 61.84 | 61.79 |
| LINEAR | narrow low band | [0.25–1] | 61.49 | 61.34 |
| | wider band | [0.25–4] | 61.51 | 61.30 |
| | wider high band | [0.25–10] | 61.42 | 60.97 |
| | shifted upward | [0.5–2] | 60.69 | 61.36 |
| | shifted downward | [0.125–2] | 61.75 | 61.54 |

