# OpenReview forum: "WaveSSM: Multiscale State-Space Models for Non-stationary Signal Attention"
_ICML.cc/2026/Conference — ICML 2026 regular_

### Official Review · Reviewer_u95N · 2026-03-12

**Soundness:** 3
**Presentation:** 4
**Significance:** 3
**Originality:** 4
**Overall Recommendation:** 4
**Confidence:** 3

**Summary:**

This paper introduces WaveSSM, a wavelet-based mechanism designed to capture multi-scale and time-localized features within Deep State Space Models (SSMs). The core contribution involves replacing the global orthogonal bases typically used in HiPPO with localized wavelet frames, which better enables improved learning from non-stationary sequence data. To support this architecture, the authors provide a formal stability analysis of the proposed frame operator and introduce a frame tightening approach to improve its numerical conditioning. The effectiveness of WaveSSM is evaluated across a broad range of benchmarks, including a newly proposed continuous copy task, as well as various classification and forecasting datasets.

**Compliance With Llm Reviewing Policy:**

Affirmed.

**Final Justification:**

This paper is a straightforward yet original combination of wavelet analysis and deep structured SSMs, representing an interesting approach to merging signal processing tools with modern deep learning architectures. My main concern is that the evaluations were quite scattered and provided little insight into why WaveSSM performs better in some tasks and worse in others.

During the rebuttal, it became clear that choosing the appropriate wavelet atoms given a fixed parameter budget is a core design decision that significantly impacts performance, and can cause WaveSSM to fail where its predecessor S4 succeeds. The authors did not adequately explore how to choose these atoms, and it remains unclear when one should use WaveSSM instead of other models in real-world scenarios. Because the rebuttal did not address these concerns, I maintain my score.

**Key Questions For Authors:**

- How does the "attention-like" mechanism in WaveSSM provide a concrete advantage over the standard Transformer architecture, particularly in capturing non-stationary dynamics?
- Could you provide a detailed runtime and memory complexity analysis of WaveSSM compared to both vanilla Transformers and standard SSMs (like S4 or Mamba)?
- In Experiment 6.3. when rescaling the input data, did you also rescale the model's underlying discretization to match the new sampling rate? If so, what was the specific methodology to rescale the model?
- What is your intuition on why WaveSSM doesn't scale to the PathX benchmark? What are the limitations in theory or practice that prevent learning from being possible?
- How do you determine which wavelet bases to use?

**Limitations:**

yes

**Strengths And Weaknesses:**

Strengths
- Well written and easy to follow.
- Good description of related works and prelimiinaries.
- Thorough theoretical analysis of Wavelet frames and also limitations of HiPPO kernels
- Extensive experiments on real data.

Weakness
- Long range arena results can be concerning, suggests that method may be insufficient to achieve original goal of modeling long-distance dependencies.
- Baselines could be improved
    - The continuous copy task only considers two baselines. Can you provide more baselines that you used in other experiments in this paper such as Mamba, and MM-SSM?
    - The baselines used for univariate forecasting do not represent the current state of the art within the forecasting literature. I recommend the authors add some baselines that are more relevant in that field. (i.e. DLinear, NHITS, TimeMixer)
- Unclear how must computational cost this incurs compared to normal S4 models.

---

> ### Author Rebuttal · Authors · 2026-03-27
>
> Dear reviewer thank you for your time and valuable comments.
>
> > Q #1: How does the "attention-like" mechanism in WaveSSM provide a concrete advantage over the standard Transformers, particularly in capturing non-stationary dynamics?
>
> For long-range sequence modeling, as tested on LRA, WaveSSM already shows an advantage over the standard Transformer baseline. We believe this is primarily because the SSM architecture provides a more suitable inductive bias for capturing long-range interactions, whereas standard Transformers do not explicitly impose a structured latent memory, which can make such dependencies harder to learn efficiently.
>
> More specifically, the advantage of WaveSSM does not replace the full flexibility of Transformer attention, but rather introduces a form of temporally localized and multiscale selectivity inside the SSMs. That said, if the goal is to assess whether this mechanism can enhance modern attention-based architectures, then comparison against more recent hybrid Transformer/Mamba models would indeed be necessary. In modalities such as language, where token-level attention is crucial, still Transformers are key element, a competitive implementation would likely require integrating WaveSSM into a hybrid Mamba/Transformer framework.
>
> > Q #2: Could you provide a detailed runtime and memory complexity analysis of WaveSSM compared to both vanilla Transformers and standard SSMs (like S4 or Mamba)?
>
> Transformer requires all-pairs token interactions, so its compute and attention memory scale as O(L^2) with sequence length. In contrast, structured SSMs provide much more efficient processing for long contexts. In particular, S4 typically uses an FFT-convolution implementation, giving a training cost of O(LlogL). Concretely, WaveSSM is implemented over S4 blocks so it has essentially the same runtime and memory scaling as S4 at training and inference time, because the main change is in the state construction and initialization, not in the sequence-processing algorithm itself. In other words, relative to vanilla S4, WaveSSM does not introduce additional running cost. Its overhead is mainly in the one-time construction of the state matrices that ocurrs only ONCE at initialization.
>
> > Q #3: In Experiment 6.3. when rescaling the input data, did you also rescale the model's underlying discretization to match the new sampling rate? If so, what was the specific methodology to rescale the model?
>
> Not at all. This experiment was designed to measure OOD generalization, following the protocol introduced in the original S4 work. The models were trained on raw 16 kHz Speech Commands and then evaluated zero-shot on undersampled 8 kHz signals, without modifying the architecture or retraining the discretization parameters. In particular, we did not apply any separate correction to the discretization step Delta, nor did we remap the continuous-time dynamics to a new sampling interval. This was intentional, since the goal of the experiment was to test robustness under a sampling-rate shift.
>
> > Q #4: What is your intuition on why WaveSSM doesn't scale to the PathX benchmark? What are the limitations in theory or practice that prevent learning from being possible?
>
> We do not observe a clear theoretical limitation showing that WaveSSM would fail in such a long-range setting. However, as we discuss in the paper, several prior works suggest that Path-X relies on very specific learning biases that make it possible for SSMs to learn in that setup. WaveSSM is designed to endow the latent state with a multiplexed representation of different temporally localized segments of the input. When the context becomes extremely long, each wavelet atom still ends up covering a very large temporal region. If Path-X requires integrating information across extremely long contexts with highly global dependencies, then the strong locality bias of the wavelet state may become a limitation rather than an advantage. In practice, addressing this would likely require increasing the order of the system to obtain finer control over the memory representation across the full sequence. At this stage, however, we do not have a clear intuition for how large that increase would need to be to succeed on Path-X.
>
> > Q #5: How do you determine which wavelet bases to use?
>
> In Section 3.1, we provide some intuition about the properties associated with each wavelet family, since each of them induces a different bias in the resulting state representation. In practice, however, we treat the choice of wavelet basis as a hyperparameter to tune. Although our experimentation was extensive, we could not identify a clear a priori winner across all tasks. In that sense, the situation is similar to HiPPO polynomial bases: it is also difficult to know beforehand which basis will be best for a given problem. In the SSM literature, for example, Legendre Scaled and Fourier Translated have prevailed as particularly effective choices, but this was established empirically.

---

> > ### Author Rebuttal · Reviewer_u95N · 2026-04-03
> >
> > Thank you for your response. It clarified some of my initials concerns (Q2, Q5) and helped me understand your method better. It is very cool that WaveSSM introduces no additional runtime. However, there are still remaining concerns that were left unaddressed, and also some new concerns that I would like to raise.
> >
> > 1. The authors did not address weak forecasting and copy-task baselines (W2) at all.
> > 2. The response to Q3 is a bit concerning. It seems that the conclusion from Experiment 6.3 is that waveSSM generalizes worse on OOD than the basic S4 methods, highlighting a critical limitation. This raises the following questions about WaveSSM.
> >     - Is an inherent limitation of WaveSSM that the transition matrices cannot be easily rescaled?
> >     - On a related note, does the wavelet formulation come at the cost of the continuous-time formulation of standard SSMs?
> >     - Why does the wavelet formulation do worse on this task? What is the failure mode that is being exposed here?
> > 3. After reading your response to Q4, it became more obvious that a fundamental weakness of WaveSSM is the selection of the appropriate scales, as also mentioned by reviewer C5Zw (Q2). If the wrong scales/shifts are chosen, then the model will fail like it did in Path-X, whereas it succeeds in S4.  Therefore, a key missing ingredient of this work is an analysis that describes how to choose the right bases given the limited compute budget mentioned in remark 6.1.  This is a non-trivial design choice that critically impacts the performance and represents a significant barrier to practical adoption if unaddressed. To maximize the impact of WaveSSM, the paper should also address how to determine the appropriate scales/shifts of the wavelets in a data-driven way, perhaps learned from data, or through some adaptive heuristic.  The paper is well-positioned to explore this question thoroughly. For instance, the proposed continuous copy task could be extended to analyze failure modes associated with very long sequences, extreme window sizes, or a mismatch between the predefined wavelet scales and the task's required temporal resolution.

---

> > > ### Author Response · Authors · 2026-04-08
> > >
> > > Dear reviewer,
> > >
> > > Thank you for letting us know that Q2 and Q5 were successfully clarified.
> > >
> > > > The authors did not address weak forecasting and copy-task baselines (W2) at all.
> > >
> > > We have conducted the additional ablations based on your suggestions. For the **continuous copying task**, we aggregated Mamba-1 and Mamba-2 in the baselines. The results are available at: https://ibb.co/7tnyPsry. In particular, we preserved the same 2-layer configuration and used the default hidden-state size, namely d\_state=16 for Mamba-1 and d\_state=128 for Mamba-2 (matching the hidden dimension of S4 and WaveSSM). This is because increasing d\_state for Mamba-1 is computationally very expensive due to the cost of selective scanning, so we kept the default setting recommended in the original paper. Overall, we observe that Mamba-1 and Mamba-2 perform between S4 and WaveSSM on this task, that is, our proposed WaveSSM still performs best on this task.
> > >
> > > Regarding the **forecasting task**, we extended our experiments to include the more recent baselines mentioned by the reviewer, including TimeMixer (2024), DLinear (2023), and others, following the long-sequence forecasting benchmarks introduced by Autoformer (2021). We took the Multivateate setup with seq\_len=96 and label\_len=48 and trained for different pred\_len=\{96,192,336,720\}. Results are available at: https://ibb.co/fVnQ7wzM. A straightforward conclusion one could make here is that, despite both S4 and WaveSSM overpassing Informer with WaveSSM being the best of the 3, SSM-based methods lag behind current state-of-the-art models on this task. We updated this extended comparison in appendix of the main paper, although we respectfully underline that our goal was not to produce a model specifically tailored for forecasting, but rather to prove that our proposed principles of locality and multiresolution work.
> > >
> > >
> > > > It seems that the conclusion from Experiment 6.3 is that waveSSM generalizes worse on OOD than the basic S4 methods.
> > >
> > > One of the conclusions for that particular Experiment that we hoped to convey to the readers is that there are potential directions of research around WaveSSMs that are worth exploring. In particular, the performance of the method on OOD tasks. For example, on those few experiments one may be tempted to say that better the model performs on ID task, worse it may perform on OOD task, as a form of a compromise (for example, the best performing model on OOD task is the worse performing model on ID task). However, we felt that it is too premature to make such conclusions, and we postponed a detailed discussion for the future research, as we reported to the Reviewer QE3M.
> > >
> > >
> > > - Is an inherent limitation of WaveSSM that the transition matrices cannot be easily rescaled? Does the wavelet formulation come at the cost of the continuous-time formulation of standard SSMs?
> > >
> > > As we follow a continuous-time formulation of the SSM, our transition matrices can be similarly rescaled through a learnable discretization step $\Delta$, without any added difficulties. In particular, similarly to standard SSMs based on polynomial-basis to construct the continous dynamics, we present an alternative that includes the principles of locality and multiresolution. In both cases those continous dynamics are discretized by a learnable parameter $\Delta$ to define the hidden state update.
> > >
> > >
> > > > After reading your response to Q4, it became more obvious that a fundamental weakness of WaveSSM is the selection of the appropriate scales, as also mentioned by reviewer C5Zw (Q2).
> > >
> > > We actually conducted thorough ablations during the rebuttal reviewer C5Zw that confirm the heuristic rule for choosing the hyperparameters that was used in the paper. Specifically, we ablated the $n_{scale}$ and the linear or log-spaced pseudo-frequency grid $[f_{min}, f_{max}]$ from where the scaling factors are sampled. The results are accessible at: https://ibb.co/JTQgpB4. There, one may note that the method is robust to the selection of those hyperparameters. Furthermore, a general rule of thumb that we followed in the paper is to set a wide pseudo-frequency default interval to [0.25-2].

---

### Official Review · Reviewer_X4Pr · 2026-03-12

**Soundness:** 2
**Presentation:** 3
**Significance:** 2
**Originality:** 3
**Overall Recommendation:** 4
**Confidence:** 3

**Summary:**

In this work, the authors propose a new class of State-space models (SSM) models in a framework named SaFARi. Specifically, it replaces the traditional polynomial basis with a new wavelet-based SSM. The new framework also has the merit of temporal localization and multi-scale representation, while the traditional basis are all global support and enroll in the information of all the historical latent states. In the experimental results, the authors show that the proposed WaveSSM outperforms orthogonal counterparts, such as the S4 on real-world application datasets with transient temporal dynamics, including  the biosiological signals on the PTB-XL dataset and raw audio on Speech Commands datasets.

**Compliance With Llm Reviewing Policy:**

Affirmed.

**Key Questions For Authors:**

I have no more questions, for my concerns, please refer to my 'weaknesses' section.

**Limitations:**

yes.

**Strengths And Weaknesses:**

Strengths:

I think the mathematical and theoretical framework proposed in this work is interesting. The proposed SaFARi framework contains wavelet frame-induced SSMs and an S4-style DPLR parameterization for real-world efficient implementation.
*  The work is supported by analysis of numerical stability considerations, and Frame tightening and rigirous comprasion between the Linear Time-Invariant HiPPO Kernels.
* The experimentation in this work is abundent, with results on ECG Transient Morphology, time-series forecasting, and raw speech classification tasks.



Weaknesses:

* The proposed SaFARi framework is technically sound but it seems to be an engineering combination of SSM and wavelet theorys.
* Figure 1 and Figure 2 are a bit hard for me to understand the whole framework and workflow of this work.
* For the experiments, the authors mainly compare with standard S4, while is S4 still the sota baseline at this time moment?

---

> ### Author Rebuttal · Authors · 2026-03-27
>
> Dear reviewer thank you for your time and valuable comments.
>
> > Point \#1: The proposed SaFARi framework is technically sound but it seems to be an engineering combination of SSM and wavelet theory.
>
> We agree that the paper combines two established ingredients:projection-based SSMs and
> wavelet/frame theory. However, our contribution is not merely to stack them together heuristically. First, we
> use the SaFARi framework to derive the induced SSM dynamics from a general frame in a principled way,
> rather than inserting a wavelet transform as a preprocessing block (as many related works [1,2,3,4,6] discussed
> in Point #1 of Reviewer C5Zw). This yields a latent state that evolves directly in a temporally localized,
> multiscale coefficient space. Second, the paper provides accompanying theory showing why such localized
> frames are better suited than global polynomial modes for piecewise-smooth signals with discontinuities.
> Third, we analyze the resulting numerical stability issues via the conditioning of the frame operator and
> propose tightening to obtain stable dynamics. Finally, the empirical comparisons are controlled at equal
> model order. We will revise the text to make clear that our originality claim is not “wavelets + SSM” per se,
> but this specific principled formulation and analysis.
>
>
>
> > Point \#2: Figure 1 and Figure 2 are a bit hard for me to understand the whole framework and workflow of this work.
>
> Thank you for pointing this out. We agree that Figures 1 and 2 could be better explained, and we will revise the wording in the main text as well as captions in the final version to better guide the reader.
>
> In particular, Figure 1 is intended to provide an initial visual intuition for the difference between globally supported polynomial bases and localized wavelet frames. Unlike polynomial bases, wavelet atoms concentrate their support on localized temporal regions, which is the source of the temporally selective memory behavior exploited by WaveSSM to provide temporal localicity.
>
> Figure 2 is intended to summarize the full workflow of the method: the input signal is projected onto a wavelet frame, SaFARi is used to derive the corresponding continuous-time SSM dynamics, and the resulting state evolves as a compact latent representation with localized and partially addressable components. We also include the Jacobian visualization to illustrate how different input times influence different parts of the final hidden state. The resulting vector at the final time step is visualized to highlight this multiplexing effect.
>
>
> > Point \#3: For the experiments, the authors mainly compare with standard S4, while is S4 still the sota baseline at this time moment?
>
> Our primary comparison to S4 was intentional and was chosen to isolate the effect of the proposed wavelet-induced dynamics from other design choices. Since WaveSSM modifies the underlying projection-induced SSM dynamics, we wanted to keep the remaining architectural choices controlled, and S4 provides the most appropriate baseline: both methods share the same general architecture and computational cost, differing mainly in the initialization and inductive bias of the state representation. This allows us to assess whether replacing global polynomial bases with localized wavelet frames yields a benefit under equal conditions.
>
> We agree, however, that S4 is no longer the overall state of the art across sequence benchmarks. The literature has evolved from time-invariant models such as S4 toward time-varying architectures such as S6/Mamba. However, Mamba induced dynamics are input-dependent, which significantly complicates the interpretation of the interal dynamics and makes the type of controlled analysis carried out in this work much less feasible. In contrast, S4 provides a more appropriate setting to study the effect of initialization bias in isolation. While the parameters are of course further optimized during training, our main goal was precisely to validate whether the proposed initialization bias enriches the state with temporal localization, at the same computational cost.

---

> > ### Author Rebuttal · Reviewer_X4Pr · 2026-04-03
> >
> > I thank the authors’ response, which addresses some of my concerns on the novelty of the framework. I will maintain my original evaluation of the paper.

---

### Official Review · Reviewer_QE3M · 2026-03-12

**Soundness:** 4
**Presentation:** 3
**Significance:** 3
**Originality:** 3
**Overall Recommendation:** 5
**Confidence:** 3

**Summary:**

This paper introduces WaveSSM, a framework that replaces the global polynomial bases typically used in State-Space Models (SSMs) like HiPPO with localized wavelet frames. Overall I think this work is a good contribution to sequence modeling by bridging classical approximation theory with modern SSMs. The empirical results on different domains, such as physiological, time series, and audio, demonstrate its superiority and potential for wider applications.

**Compliance With Llm Reviewing Policy:**

Affirmed.

**Final Justification:**

I find this work to be original with solid experiments. While some experimental results could use further interpretation, it does not affect the overall quality of this paper. I therefore kept my original score for acceptance.

**Key Questions For Authors:**

Could authors provide some interpretation on why S4 is more robust on the speech commands task with 8khz sampling rate?

**Limitations:**

yes

**Strengths And Weaknesses:**

- Soundness: The derivation of transition matrices using the SaFARi framework to handle non-orthogonal wavelet frames is theoretically sound. The authors further show that the frame tightening procedure effectively mitigates ill-conditioning in redundant wavelet frames. The empirical analysis scope can be expanded but obtained results are strong enough and suggest its potential to scale to other sequence tasks.
- Presentation: I found the problem decomposition is clear and authors did a good job motivating their architectural design step by step.
- Significance: I do think the derivation of SSM dynamics from wavelet-frame projections is a significant departure from global polynomial traditions. Combined with its empirical results on different tasks, the proposed model could be of benefit for other sequence modelling tasks.
- Originality: The derivation of SSM dynamics from wavelet-frame projections is interesting. Creating addressable latent states offers an efficient alternative to Transformer-style attention.

---

> ### Author Rebuttal · Authors · 2026-03-27
>
> Dear reviewer, thank you for your time and valuable comments. We agree that the departure from global polynomial is not trivial, especially when comparing our sparse solution under the exact same computation constraints. This opens new directions to explore sparser transition dynamics that are comparable or, as shown in this work, even more performant for certain tasks.
>
> > Q #1: “Could authors provide some interpretation on why S4 is more robust on the speech commands task with 8khz sampling rate?”
>
> This question is very profound. We speculate that resampling may have a stronger effect on WaveSSM because changing the sampling rate compresses the temporal dimension of the input, which may cause a severe shift in the internal modes or atoms activated by the SSM. In WaveSSM there is a direct relation between the active modes and their locations in the input signal. Mitigating this effect is not trivial and it would be a good direction to explore by itself.
>
> Another plausible explanation is that, due to the greater expressivity of the framework, WaveSSM may overfit more closely to the training distribution at 16 kHz, so that evaluation at 8 kHz effectively becomes a more severe out-of-distribution shift.

---

> > ### Author Rebuttal · Reviewer_QE3M · 2026-04-02
> >
> > I appreciate authors' explanation on this. While I do hope for a deeper investigation of this, I believe it is out of the scope of the current manuscript and can probably be left for future investigation. I will keep my score as it is as I believe it reflects the good quality of this work.

---

### Official Review · Reviewer_C5Zw · 2026-03-14

**Soundness:** 2
**Presentation:** 3
**Significance:** 2
**Originality:** 3
**Overall Recommendation:** 4
**Confidence:** 3

**Summary:**

This paper introduces a novel wavelet-based state-space model (waveSSM) that integrates wavelet frames into the construction of SSM dynamics based on the SaFARi framework. Unlike traditional SSMs such as HiPPO and S4 that rely on global polynomial bases (e.g., Legendre), waveSSM leverages the time-frequency localization properties of wavelets to better capture non-stationary and transient structures in sequential data such as physiological signals, time series, and audio. Furthermore, this paper investigates the numerical stability and approximation precision of the proposed method for theoretical validity, and demonstrates the superior accuracy and performance through extensive experiments on both synthetic benchmarks and real-world datasets.

**Compliance With Llm Reviewing Policy:**

Affirmed.

**Final Justification:**

The authors have addressed my concerns properly.

**Key Questions For Authors:**

Q1: The waveSSM demonstrates strong performance on continuous, non-stationary signals due to its time-frequency localization bias. However, a dominant application area of modern SSMs (such as Mamba and S4) is discrete symbolic modalities, particularly language modeling, which relies on capturing non-local semantic dependencies. Given the inherent locality of wavelet bases, do you anticipate that waveSSM can be directly applied to discrete token sequences, or would it require fundamental architectural modifications to handle the long-range, hierarchical dependencies prevalent in language?

Q2: The construction of the wavelet frame involves several critical hyperparameters, including the number of scales n_scales, the translation density factor alpha, and the choice of the mother wavelet (e.g., Morlet vs. Gaussian derivatives). While the paper provides empirical ranges for these values (e.g., n_scales ∈[24,64] in Appendix C.5), there is no systematic ablation study on their sensitivity. Could you qualitatively discuss how varying these hyperparameters (e.g., using too few scales, or a density factor alpha that is too large) would impact the model‘s ability to capture transient dynamics and its numerical stability? Are there diagnostic tools to guide the optimal selection of these parameters for a new dataset?

Q3: Theorem B.1 provides a theoretical guarantee that wavelet frames achieve faster
approximation rates than polynomial bases for piecewise smooth signals, relying on the Parseval frame assumption. In practice, the constructed discrete frames are only approximately Parseval. Have you conducted any empirical diagnostics to measure how "tight" the constructed frames are (e.g., computing the condition number of the Gram matrix)? Is there a correlation between the "tightness" of the frame and the final task performance across different datasets?

**Limitations:**

Yes

**Strengths And Weaknesses:**

Soundness:
	The paper is built on a solid mathematical foundation, such as SSM frame theory, wavelet analysis, and approximation theory. The theoretical results (e.g., Theorem B.1) rigorously demonstrate the superiority of wavelet frames over polynomial bases for signals with discontinuities. Besides, the authors proactively address numerical stability that is a critical issue in wavelet-induced dynamics. They analyze the condition number of the Gram matrix and its impact on the system matrix A (Appendix D.2), providing design guidelines to ensure stable long-range kernels. Moreover, the experimental setup emphasizes controlled comparisons. For instance, in Remark 6.1, the authors use the same model order as polynomial-based baselines to perform a fair comparison, which strengthens the validity of the reported performance gains.

Presentation:
	This paper is well-structured and clearly motivated.

Significance:
	This paper  successfully integrates a classical signal processing tool (wavelets) into modern SSM architectures.

Originality:
	The use of wavelets in constructing state-space models (SSMs) is not entirely novel. Several studies have already explored the combination of wavelets and SSMs.

Weaknesses:
There are two potential weaknesses that warrant further investigation or refinement. First, the theoretical analysis relies on the assumption of Parseval frame. However, the discrete wavelet frames constructed in practice only approximately satisfy this condition. The paper does not quantify the impact of this approximation error on model performance or stability. Second, there is a lack of ablation studies or sensitivity analysis on key hyperparameters, such as the number of scales, the density of translations, and the choice of mother wavelet. This makes it difficult to assess the robustness of the method to parameter tuning.

---

> ### Author Rebuttal · Authors · 2026-03-30
>
> Dear reviewer thank you for your thoughtful and valuable comments.
> >Point \#1: The use of wavelets in constructing SSMs is not entirely novel. Several studies have already explored the combination of wavelets and SSMs.
>
> We agree that the combination of wavelets and SSMs has recently attracted growing attention. Although most of that works rely on wavelet transforms as a feature decomposition or preprocessing step, rather than defining the dynamics of the SSM. We provide a brief summary here, while the full comparison will be included in the paper. Previous works as [5], proposes a multiscale dyadic decomposition based on DWT implemented as a cascade of conv1d with increasing dilation across scales; each scale is then processed by a separate SSM. The method is independent of SSM implementation so they try in S4 and S6 Mamba blocks. Similarly, [3] uses DWT to split the input into multiple frequency bands and in time series [2] aimed at designing adaptive wavelet preprocessings per channel, or anomaly detection [1].
>
> Furthermore, our work significantly differs and deepens [4]. In particular, we provide the accompanying theory showing why localized frames are better suited than global polynomial modes for piecewise-smooth signals with discontinuities. We analyze the resulting numerical stability issues via the conditioning of the frame operator and propose tightening to obtain stable dynamics. Finally, we provide empirical comparisons at controlled environments in order to assess the benefits of the proposal.
>
> [1] Wave-MambaAD, ICCV (2025)
> [2] WaveState, Knowledge-Based Systems (2025)
> [3] STEAM, ACM (2025)
> [4] WaLRUS, NeurIPS (2025)
> [5] MS-SSM, ICLR Workshop (2025)
>
> >Q \#1: Given the locality of wavelet bases, do you anticipate that waveSSM can be applied to handle the long-range, hierarchical dependencies prevalent in language?
>
> We agree that WaveSSM has a stronger locality bias than polynomial-based SSMs, which is why we focus on non-stationary and transient signals. However, this locality is multiscale, not purely short-range: coarse scales provide broader temporal support. In addition, in a deep architecture, stacked layers can compose localized features into more global representations. Empirically, this is also why we view the approach as promising beyond purely continuous signals: on LRA, WaveSSM is competitive on symbolic/long-context tasks and improves over S4 on 4 of 6 tasks. That said, for discrete token sequences, we believe the current formulation is promising but likely not yet sufficient to fully capture the highly non-local semantic dependencies that strong language models require at token-level granularity. Our intuition is that language modeling requires stronger context-dependent interactions, such as token-level attention in Transformers or the per-token selective dynamics of Mamba. In that setting, additional architectural modifications would likely be necessary. We see this as an important direction for future work.
>
> >Q \#2: Could you discuss how varying these hyperparameters (scales, or a density factor alpha) would impact the model‘s ability to capture transient dynamics and its numerical stability?
>
> Thank you for raising this point. Apart from the choice of the mother wavelet ablated in the main text, we have conducted complementary ablations always having in mind the Remark 6.1 and preserving the "same budget" in all cases. Particularly, we ablate the n_scale and the linear or log-spaced pseudo-frequency grid [f_min, f_max] from where the scaling factors are sampled. The experiments conducted on MorletS and MorletT wavelets on Cifar and Listops datasets are accessible at: https://ibb.co/JTQgpB4. Results show the method is robust to the selection of those hyperparameters. A general rule of thumb that is robust enough and that we followed in the paper, is to set $n_{scale} := \sqrt{N}$, and set a wide pseudo-frequency default interval to [0.25-2] (and choose from it based on $n_{scale}$).
>
> >Q \#3: The constructed frames are only approximately Parseval. Have you measured how "tight" the constructed frames are (e.g., cond. number of the Gram matrix)? Is there a correlation between the "tightness" of the frame and performance?
>
> In practice, before tightening the frame, we experienced stability issues on most datasets. This was identified early in the development of the paper, and frame tightening became a necessary step to make the proposed construction numerically stable. As a result, all experiments reported in the paper use the tightened frame. Therefore, rather than being merely correlated with improved performance, tightening was in many cases a required condition for stable convergence of the method. We will clarify this point more explicitly in the revised version of the paper.

---

> > ### Author Rebuttal · Reviewer_C5Zw · 2026-04-05
> >
> > The authors have addressed my concerns properly. I have adjusted my score accordingly.

---

### Decision · Program_Chairs · 2026-04-30

**Decision:**

Accept (regular)

**Comment:**

The paper introduces waveSSM, a wavelet-based state-space model. The reviewers highlighted the novelty and extensive empirical evaluations. All major issues have been successfully addressed during the rebuttal. I recommend acceptance.